# The 3 × 2 Achievement Goals in the Education, Sport, and Occupation Literatures: A Systematic Review with Meta-Analysis

Marc Lochbaum [1,2,*], Cassandra Sisneros [1] and Zişan Kazak [3]

1 Department of Kinesiology and Sport Management, Texas Tech University, Lubbock, TX 79409, USA; cassandra.sisneros@ttu.edu
2 Education Academy, Vytautas Magnus University, 44248 Kaunas, Lithuania
3 Faculty of Sport Science, Ege University, Bornova, Izmir 35100, Turkey; f.zisan.kazak@ege.edu.tr
* Correspondence: marc.lochbaum@ttu.edu

**Abstract:** Achievement goal theory has been a dominant motivation framework since the 1980s. The 3 × 2 achievement goal framework emerged in the literature in 2011. We aimed to conduct a systematic review with meta-analysis following the PRISMA guidelines of the 3 × 2 achievement goal research in education, sport, and occupation settings. We retrieved articles from searching EBSCOhost and Google Scholar platforms. Eligible articles contained the 3 × 2 achievement goal in education, sport, or occupation, were published in a peer-reviewed journal, and provided mean data or correlate data. We tested hypotheses concerned with (1) the overall pattern of achievement goal endorsement, (2) achievement goal differences by domain (education, sport) and compulsory nature of the domains or sub-domains, and (3) achievement goal relationships with correlates (e.g., learning strategies, motivations, performance). After screening, 56 articles met all inclusion criteria, providing 58 samples across education ($n = 44$), sport ($n = 10$), and occupation ($n = 4$) settings with 35,031 unique participants from 15 countries. Participants endorsed the task- and self-approach goals more than the counterpart avoidance goals, other-avoidance goals more than other-approach goals, and the intercorrelations and reliability coefficients were acceptable. Minimal impact results from examining within and across study bias statistics. Of importance, the domain (i.e., education, sport) and the compulsory nature of the domain or sub-domains (i.e., primary-secondary education, sport) moderated goal endorsement (group mixed-effects $p < 0.05$, $g$ values medium to very large). These groupings also moderated the other goal differences. Concerning our correlates analyses, most meta-analyzed correlations among the achievement goals and correlates were small in meaningfulness with the largest correlations (0.30–0.42) between the approach goals merged and the task- and self-approach goals and facilitative learning strategies and desired motivations. In conclusion, the 3 × 2 achievement goals literature is diverse. Furthering the study and application of this model requires overcoming inherent limitations (i.e., consistent response scale sets), teasing out differences between the task- and self-goals, measuring performance outcomes, and cross-cultural collaborations.

**Keywords:** motivation; approach goals; avoidance goals; task goals; self-goals; other goals; quantitative review

## 1. Introduction

Since the late 1970s, researchers have embraced the study of achievement goal theories in the achievement motivation literature [1–5]. From initial dichotomous achievement goal frameworks [6–8], a number of other frameworks appeared in the literature [2,3,9]. Researchers took to each new framework, such as the trichotomous and the 2 × 2 achievement goal frameworks, that meta-analyses introduced in the literature [10–15]. The most recent framework with both goal definition and valence is Elliot, Murayama, and Pekrun's 3 × 2 model [9]. Unlike the other frameworks, a systematic review is absent. Hence, we aimed to provide a quantitative review of the literature in the education, sport, and occupation domains.

### 1.1. The 3 × 2 Achievement Goal Model

The initial dichotomous framework included mastery or task and performance or ego as competence measures [6–8]. Elliot and colleagues first bifurcated the performance goal by incorporating the approach–avoidance distinction [4]. From his trichotomous achievement goal framework, Elliot expanded the approach–avoidance distinction to the mastery goal. Then, in 2011, Elliot and colleagues [9,12] proposed their 3 × 2 achievement goal model. To expand upon the 2 × 2 achievement goal framework, Elliot et al. divided the mastery component of the 2 × 2 framework into task-based and self-based goals. This separation allowed for the comparison of process-oriented and intrapersonal goals. The performance approach and avoidance goals followed the basic definitions found in the 2 × 2 achievement goal framework (i.e., focus on the attainment of other based competence or avoidance of other based incompetence). Hence, the 3 × 2 achievement goal model includes three goals through which individuals in achievement contexts may choose to define their competence: task, self, and other [9]. These three standards of competence may be positive or negative in valence depending on whether motives are approaching success or avoiding failure.

Concerning each achievement goal, Elliot, Murayama et al. [9] conceptualized the task-based component, which uses the demands of the specified task as the evaluative referent. Thus, how well one fulfills what the task requires measures competence and success in the task-based component. In this process-oriented goal component, one may approach the task with the intention of carrying the task out correctly or avoiding doing the task incorrectly. These two valences make up the sub-components task approach and task avoidance. The self-based, or intrapersonal, component uses one's intrapersonal trajectory as the evaluative referent. Competence and progress in self-based goals are related to one's past performance or to one's future potential. The difference in motives is what differentiates self-approach and self-avoidance; self-approach is striving to outperform one's past performance, while self-avoidance is striving to avoid an inferior performance than before. Elliot, Murayama et al. [9] highlighted the differences between self-approach and self-avoidance as predictors of scholastic behavior in their antecedent analysis. Self-approach goals, in their initial research, were positive predictors of energy in class, while self-avoidance goals had a near equal yet opposite effect.

The other based, or interpersonal, component uses others' performance as the evaluative referent and uses interpersonal comparison for motivation. The aim of other-based goals is to generate feelings of confidence or to avoid feelings of shame. The difference in intent of interpersonal comparison distinguishes other-approach and other-avoidance. Throughout life, with an emphasis during the primary years, individuals within achievement motivation contexts often compare themselves with others to gauge competence. Other-based competence is more evident in compulsory-related activities than voluntary. An example setting that distinguishes other approach versus other avoidance is a physical education class. A student may strive to perform in a superior way compared to peers, generating feelings of confidence and efficacy, or they may focus on performing well to avoid the feelings of shame and embarrassment that are associated with an inferior performance.

The 3 × 2 model's applicability fostered the creation of a number and wide variety of scales. In the educational domain exists the Social Studies Oriented Achievement Goal Scale [16], the Pictorial Achievement Goal Measurement for Kindergarteners [17], Questionnaire on Teamwork Learning Goals [18], and the 3 × 2 Achievement Goal Questionnaire for Teachers [19,20]. In the sport domain, variations include the 3 × 2 Achievement Goal Questionnaire (AGQ) for Sport [21], AGQ for Physical Education [22], and AGQ for Recreational Sport [23]. Other revisions to Elliot's original 3 × 2 questionnaire include language translations [24,25] and changes in phrasing for relevant application to the study focus, such as work or homework [26,27].

### 1.2. Purpose and Hypotheses

To date, no reviews, of any kind, exist on Elliot and colleagues' 3 × 2 achievement goal model. We sought to carry out a meta-analysis on this achievement goal model

across education, sport, and occupation literatures. We planned to assess domain and the compulsory nature differences and correlates of the 3 × 2 achievement goals. To this end, we assessed the following hypotheses.

**Hypothesis 1.** *Our first hypothesis concerned the overall pattern of achievement goal endorsement. We hypothesized, based on Elliot and colleagues' [9] data and Lochbaum et al.'s [15] sport-based meta-analysis, that participants will endorse the task- and self-approach goals more than the counterpart task- and self-avoidance goals and will endorse the task- and self-goals more than the other goals. Elliot's research suggests participants endorsed the other-avoidance goals more than the other-approach goals, at least in their higher education samples. Given we expected data with both 5- and 7-point response scales, we hypothesized the same overall patterns. Within this first hypothesis, we examined the intercorrelation among the achievement goals and reliability coefficients. Again, based on Elliot and colleagues' [9] and Lochbaum et al.'s [15] works, we hypothesized a range of intercorrelations, with most being moderate in meaningfulness (i.e., r range 0.50–0.79) and with acceptable (i.e., >0.70) reliability coefficients.*

**Hypothesis 2.** *Our second hypothesis concerned differences in the overall pattern of achievement goal endorsement based on domain, education or sport, and the compulsory nature of domain (primary and secondary education as compulsory, and sport and higher education as non-compulsory). We hypothesized, based on Lochbaum et al.'s [15] meta-analysis that examined PE versus sport and leisure-time physical activity, that all goals will be endorsed more in sport than in education, and in non-compulsory activities compared to compulsory ones. The differences we hypothesized will be more pronounced within the compulsory and non-compulsory analyses. Last, within moderator groups, we sought to examine more in depth the pattern of other-avoidance goal endorsement such as the difference between PE and sport samples. Lochbaum and colleagues' [14] meta-analysis suggests we should expect PE samples to have higher other-avoidance goals than sport samples.*

**Hypothesis 3.** *Our last hypothesis concerned the relationships between correlates such as self-determined motivations, affect, and achievement strategies and outcomes according to the achievement goal valence and each of the six achievement goals. We hypothesized, based on Elliot and colleagues' [9] and all of Lochbaum and colleagues' [13–15] meta-analyses, the approach goals to be most related to our outcomes compared to the avoidance goals. We expected most relationships to be significant, yet small in meaningfulness.*

## 2. Materials and Methods

This systematic review with meta-analysis followed the PRISMA guidelines [28]. Though we did not register our protocol, we checked the PROSPERO database to ensure we were not replicating a similar review prior to starting our review.

### 2.1. Eligibility Criteria and Selection

For inclusion purposes, the articles met the following criteria: (a) article of any research design published in a peer-review journal from 2011 (date of Elliot and colleagues' [9] publication of the 3 × 2 achievement goal model until the end of our search); (b) a 3 × 2 achievement goal measure; (c) participants, anticipated as ranging from elementary school children to adults, engaged in education, sport, or occupation domains during the time of questionnaire completion; and (d) sufficient data (i.e., 10 samples) for analyses to be tested in at least one of our hypotheses to avoid misleading results [29]. Though we did not impose a language of publication restriction, we conducted our main search in English; hence, only articles in any language with corresponding titles, abstracts, and keywords in English resulted from our search. Based on the initial findings and one author's native language being Turkish, hand-searching was carried out in the Turkish language.

## 2.2. Information Sources, Search Strategy, and Search Protocol

First, C.S. conducted the search in Google Scholar and EBSCOhost (i.e., APA PsycINFO, ERIC, and Psychology and Behavioral Sciences Collection). Both the Google Scholar and EBSCOhost searches began on 17 January 2023, and ended 7 February 2023. C.S. used the following search terms on Google Scholar: "3 × 2 achievement goal model" and achievement goal model and sport. In EBSCOhost, we used 3 × 2 achievement goal model or 3 × 2 achievement goal* or AGQ-S. M.L., and Z.K. examined the main search results, which was completed by C.S. Next, M.L. and C.S. together then completed hand-searching (see Figure 1). Last, after examining the search and noticing the number of articles in Turkish, Z.K. completed her hand-searching process in her native Turkish language. All search details are located in our Supplementary Materials and represented in Figure 2. Discrepancies regarding thoughts of inclusion or exclusion were handled by M.L. and C.S. discussing the articles under question.

Definition

|  |  | Absolute (task) | Intrapersonal (self) | Interpersonal (other) |
|---|---|---|---|---|
| Valence | Positive (approaching success) | Task-approach goal | Self-approach goal | Other-approach goal |
|  | Negative (avoiding failure) | Task-avoidance goal | Self-avoidance goal | Other-avoidance goal |

**Figure 1.** The 3 × 2 achievement goal model. The definition and valence dimensions represent competence. The absolute, intrapersonal, and interpersonal definitions are three ways to define competence. Competence may be valenced as either positive or negative.

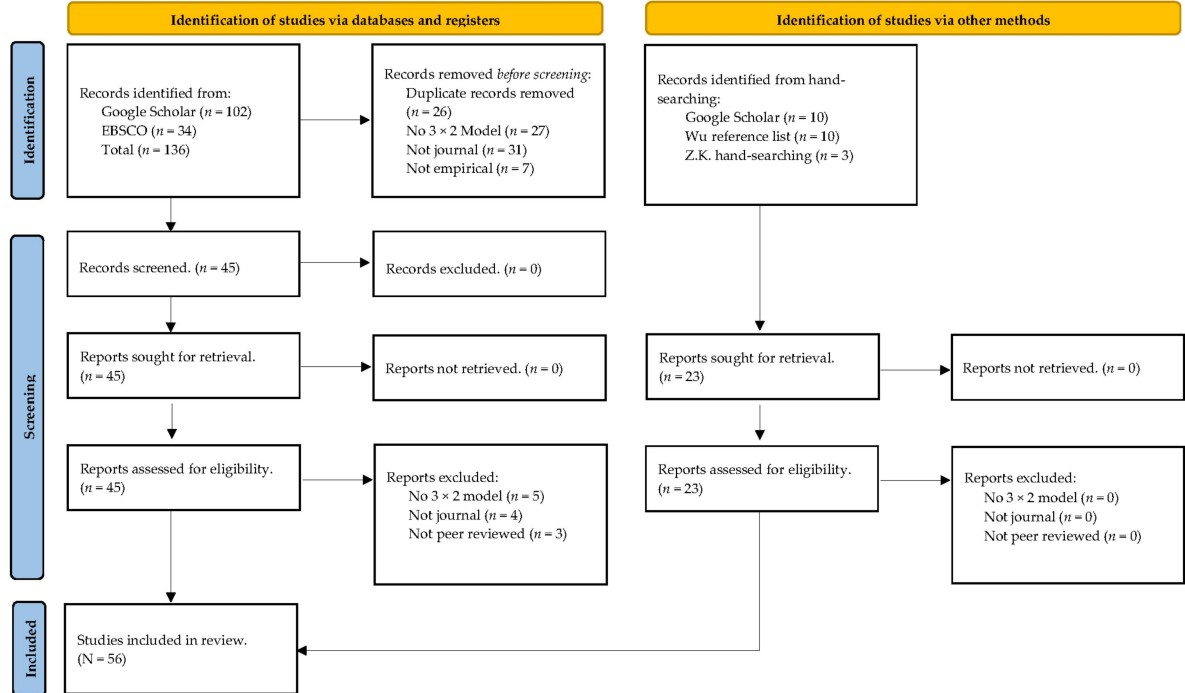

**Figure 2.** Search strategy.

### 2.3. Data Collection and Items Retrieved

Together, M.L. and C.S. extracted the following information: reference, sample size, sample percent male, country of participants, age mean, range or description, domain and sub-domain, the 3 × 2 Likert scale range, the 3 × 2 mean, intercorrelations, reliability coefficients, and the correlate scale domain and correlation values with the 3 × 2 achievement goals. We sought clarification from seven authors and received three replies. For the Turkish literature, Z.K. a native speaker, located all of the required information after confirming the studies met our inclusion criteria. For the non-English 3 × 2 achievement goal literature, including Spanish, Russian, and Chinese, we used Google Translate (https://translate.google.com/, last accessed 10 March 2023).

### 2.4. Risk of Bias Assessments

If possible (i.e., number of studies, differences in risk ratings), we planned to examine whether either risk moderated our mean level and correlate results. As found in Table 1, we coded and scored the studies on six individual study risks of bias. For risk of bias across the studies, we examined publication bias with the classic fail–safe *n*, Orwin's fail–safe *n*, the funnel plot, and the 'trim and fill' results. The classic fail–safe *n* statistic represents the number of null samples required to change a significant value into a non-significant value [30]. Unlike the classic fail–safe *n*, Orwin's fail–safe *n* is the number of missed studies that, when added to the actual data, would move the new correlation past a chosen threshold [31]. We chose $r = 0.00$ as our missed study value, and $r = 0.10$ as our threshold due to this value being the lower end of a correlation with low meaningfulness. Hence, the greater the value for the classic and Orwin's fail–safe *n* calculations, the greater the confidence and thus robustness of the results. We specified the one-tail test when we conducted the classic fail–safe *n* analysis. As funnel plots represent whether the retrieved studies capture the essence of all studies, we examined whether the entered studies dispersed equally on either side of the overall effect [32]. Last for our planned risk of bias analyses, we used Duval and Tweedie's [33] trim and fill analysis. When required, data points filled to the right increase the point estimate regardless of it being the 3 × 2 achievement goal means or examined correlations, whereas those filled to the left lower the point estimate.

**Table 1.** Risk of bias categories and coding information.

| Bias | Low Risk (3 Point) | Medium Risk (2 Points) | High Risk (1 Points) |
|---|---|---|---|
| Sample selection | Sampling across multiple groups | Sampling within larger group | One convenience sample |
| Nonresponse | Appears most participants completed the measures | Some nonresponse occurred | Seems most did not do it |
| Collection | In person | Mix | All online or mail or not stated |
| Collection method | All the same | | Not the same or not stated |
| Anonymity | Stated | IRB approval | Not stated |
| AGQ measure | Elliot [9] or slight adaptation | Adapted variation (e.g., language) | Based on Elliot [9] |

### 2.5. Summary Statistics, Planned Analyses and Certainty Assessment

To test our hypothesis pertaining to the 3 × 2 achievement goal scores, we entered the mean values. For our mean score statistics, we calculated Hedges' g and followed standard guidelines, with 0.20 as small, 0.50 as medium, 0.80 as large, and 1.30 as very large [34]. To evaluate our relationship hypothesis among the 3 × 2 achievement goals and correlates, we entered correlation coefficients. We interpreted the correlation values as follows: 0.10–0.29 as small, 0.30–0.49 as medium, and 0.50 or greater as large. We assumed heterogeneity in all our analyses, as heterogeneity exists in sport psychology meta-analyses [35]. Thus, we planned to conduct and report both random- and mixed-effects analyses. For our statistics, we reported the number of cases, sample size, point estimate (i.e., effect size), 95% confidence intervals, heterogeneity, and publication bias statistics. We reported the $I^2$ statistic, the ratio of excess dispersion to total dispersion, as our heterogeneity measure with the following interpretation: <25 (low), at least 50 (medium), and >75 (high) [36]. To

examine our moderator-based hypothesis, we used a mixed-effects analysis. For these analyses, we reported the number of cases, sample size, point estimate (i.e., effect size), 95% confidence intervals, and the Q total between (QTB) with an associated *p*-value. The QTB indicates the level of difference between different moderator levels. Though we anticipated our analyses to be underpowered, we set the statistical significance at the traditional $p < 0.05$. We conducted our meta-analyses using the Comprehensive Meta-Analysis (CMA) version 3 software (version 3.3.070, Biostat, Inc., Englewood, NJ, USA, 20 November 2014) and ran our descriptive analyses with Intellectus Statistics (https://www.intellectusstatistics.com/). To evaluate certainty, we examined our reported statistics such as confidence intervals, risk of bias statistics, and differences between groups and correlations.

## 3. Results

### 3.1. Study Selection and Characteristics

Based on our PRISMA search strategy (see Figure 1), 56 studies met all inclusion criteria from 2011 until 2022, with 32 studies from 2011 until 2019 and the remaining 24 studies from 2020 until 2022. Table 2 contains information concerning the study reference, number of participants, country of the participants, mean sample age or information provided, percentage of male participants, the domain (i.e., education, sport, or occupation), the sub domain (e.g., higher education, specific sport), the achievement goal measure name and response set. The samples totaled 35,031 participants (493.39 ± 516.39, range 12 to 2630) originating from 15 countries, with 24.6% from Spain, 15.8% from Turkey, and 14.0% from the United States of America as the most represented nationalities. Of the 58 included samples, 44 were education, 10 were sport, and 4 were occupation. Participants ranged from kindergarten students to working adults, with 40% of participants being under 18. Our studies included both males and females with 45.61% (SD = 18.63%) of participants identifying as male; the minimum percentage of male study participation was 10%, while the maximum percentage was 100%. Over 50% of the included studies used a 7-point Likert scale, with 5-point scale use following at 35%. The remaining studies used a 4- or 6-point Likert scale.

### 3.2. Risk of Bias within Studies

Figure 3 contains the risk of bias within studies information. The minimum possible quality score was 6.00, while the maximum possible quality score was 18.00. The mean quality score of samples across all risk categories was 13.96 [95% CI 12.99, 14.94] and the median quality score was 15.00. Dividing the scores into three groups, a score of 14.00 or below was considered low quality, studies scoring 15.00 were considered medium quality, and studies scoring 16.00 or greater were considered high quality. We compared quality scores between study domains (education, work, and sport) as well as compulsory nature of domains. These comparisons yielded negligible differences between the groups.

### 3.3. Hypothesis 1 Results

Our first hypothesis concerned the overall pattern of achievement goal endorsement, intercorrelations, and reliability coefficients. Table 3 contains the intercorrelations, both fixed and random effects, and averaged reliability coefficients. With the fixed-effects results, the intercorrelations were ($n = 1$) medium ($n = 6$) and large ($n = 8$) in meaningfulness and these results shifted to a couple smaller ($n = 3$) intercorrelations with the random effects model. The reliability coefficients were acceptable (i.e., >0.70) with all being greater than 0.80, and the 95% confidence interval lower limits being below 0.80 only for the task and self-avoidance achievement goals.

As found in Tables 4 and 5, we examined the achievement goals at individual subscale level and provided mean (fixed and random values), heterogeneity, and publication bias statistics. Across all studies with both the 7- (Table 4) or 5-point (Table 5) rating scales, the participants endorsed the task goals more than the self and other goals. The participants endorsed the task and self-approach goals more than the counterpart avoidance goals.

However, the participants endorsed, on average, the other avoidance goals more than the other approach goals. Heterogeneity was high for each of the achievement goals. Concerning bias (see Figure 4 for the 7-point Likert scale funnel plot and Figure 5 for the 5-point Likert scale funnel plot), a slight downturn results for most goals, suggesting lower rated achievement goal data are missing from the published literature. Given the chance of the mean value lower 95% confidence interval crossing zero requires extreme variability and a much lower mean score, the fail–safe *n* values were all large (greater than 1 million). Our table indicates greater than 1000 as sufficient for this statistic.

**Table 2.** Study information, participants characteristics, and 3 × 2 achievement goal information for all studies contributing to at least one analysis.

| | Participant Characteristics | | | | | Scale Characteristics | | |
|---|---|---|---|---|---|---|---|---|
| Study | N | Country | Age | % Male | Domain | Sub Domain | AGQ | Likert |
| Agbuga [37] | 303 | TR | 21.51 | 56 | ED | HED | ? | 7 |
| Alasqah [38] | 149 | SA | HED | 34 | ED | HED | AGQ | 5 |
| Cecchini et al. [39] | 334 | ES | 13.12 | 53 | ED | PE | 3 × 2 AGQ-PE | 5 |
| Cetin [40] | 658 | TR | 20.21 | 26 | ED | HED | ? | 7 |
| Chung-Chin [41] | 275, 252 | TW | 13.00, 11.00 | 53, 46 | ED | SEC, PRI | ? | ? |
| Cowden et al. [42] | 323 | ZA | 17.60 | 69 | Sport | Tennis | 3 × 2 AGQ-S | 7 |
| Danthony et al. [43] | 486 | FR | 15.83 | 38 | ED | PE | 3 × 2 AGQ-S | 7 |
| Didin and Kasapoglu [44] | 440 | TR | 12.00 | 47 | ED | SEC | SS-O AGS | 5 |
| Diseth et al. [45] | 217 | NO | 22.67 | 19 | ED | HED | 3 × 2 AGQ | 7 |
| Elliot et al. [9] | 126, 319 | DE; US | HED | 18, 65 | ED | HED | 3 × 2 AGQ | 7 |
| García-Romero et al. [46] | 205 | ES | 14.02 | 55 | ED | PE | 3 × 2 AGQ-PE | 5 |
| García-Romero et al. [47] | 1706 | ES | 13.75 | 53 | ED | PE | 3 × 2 AGQ-PE | 5 |
| García-Romero et al. [48] | 1706 | ES | 13.75 | 53 | ED | PE | 3 × 2 AGQ-PE | 5 |
| Gezer and Şahin [16] | 374 | TR | SEC | 53 | ED | SEC | SS-O AGS | 5 |
| Gillet et al. [27] | 278, 327, 169 | FR | 18.93, 18.93, 32.48 | 17, 17, 44 | ED; ED; Profession | HED | 3 × 2 AGQ | 7 |
| Hidayat et al. [49] | 538 | ID | 18–22 | 10 | ED | HED | 3 × 2 AGQ | 7 |
| Hidayat et al. [50] | 538 | ID | 18–22 | 10 | ED | HED | 3 × 2 AGQ | 7 |
| Hidayat et al. [51] | 538 | ID | 18–22 | 10 | ED | HED | 3 × 2 AGQ | 7 |
| Hunsu et al. [52] | 437 | US | 20.95 | 73 | ED | HED | 3 × 2 AGQ | 5 |
| Ireri et al. [53] | 385 | KE | 16.65 | 50 | ED | SEC | AGQ | 5 |
| Kadıoglu-Akbulut and Uzuntiryaki-Kondakcı [54] | 197, 311 | TR | 20.40, 19.14 | 21, 19 | ED | HED | ? | 7 |
| Karahan [55] | 68 | TR | Adult | 38 | Profession | PRI/SEC | 3 × 2 AGQ | 7 |
| Kılıçoğlu [56] | 346 | TR | SEC | 50 | ED | SEC | SS-O AGS | 5 |
| Kovács et al. [57] | 21, 31, 47, 28 | HU | 16.16 | 51 | Sport | Karate | 3 × 2 AGQ-S | 7 |
| León-del-Barco et al. [18] | 700 | ES | 21.23 | 37 | ED | HED | QTLG | 7 |
| Liu and Liu [58] | 159 | US | HED | NR | ED | HED | 3 × 2 AGQ | 7 |
| Lower and Turner [23] | 250, 343 | US | 18–22 | 69 | Sport | IM Sport, Sport Clubs | 3 × 2 AGQ-RS | 6 |
| Lower et al. [59] | 907 | US | HED | 52 | Sport | HED | 3 × 2 AGQ-RS | 6 |
| Lower-Hoppe et al. [60] | 890 | US | 20.66 | 49 | Sport | Club, Intramural, Group fitness | 3 × 2 AGQ-RS | 6 |
| Lüftenegger [61] | 388 | AT | 25.00 | 18 | ED | HED | 3 × 2 AGQ | 7 |
| Mascret et al. [21] | 679, 302 | FR | 21.50 | 68, 71 | ED | PE | 3 × 2 AGQ-S | 7 |
| Mascret et al. [19] | 304 | FR | 38.25 | 39 | Profession | ED | 3 × 2 AGQ-teachers | 7 |
| Mascret et al. [62] | 38 | FR | 18.50 | 100 | Sport | Basketball | 3 × 2 AGQ-S | 7 |
| Méndez-Giménez et al. [22] | 150, 366 | ES | 13.97, 14.11 | 50, 49 | ED | PE | 3 × 2 AGQ-PE | 5 |
| Méndez-Giménez et al. [63] | 1347 | ES | 13.43 | 57 | ED | SEC | 3 × 2 CGSQ | 5 |
| Méndez-Giménez et al. [64] | 2630 | ES | 14.39 | 53 | ED | SEC | 3 × 2 AGQ | 7 |
| Méndez-Giménez et al. [65] | 1689 | ES | 13.25 | 51 | ED | PE | 3 × 2 AGQ-PE (Spanish) | 5 |
| Méndez-Giménez et al. [66] | 405, 646, 559 | ES | 10–12, 13–14, 15–17 | 48, 57, 53 | ED | PE | 3 × 2 AGQ-PE (Spanish) | 5 |
| Méndez-Giménez et al. [67] | 2284 | ES | 14.31 | 52 | ED | SEC | 3 × 2 AGQ-PE (Spanish) | 7 |
| Nikitskaya and Uglanova [24] | 280 | RU | 12–18 | 53 | ED | SEC | 3 × 2 AGQ (Russian) | 4 |
| Ning [68] | 384 | CN | 19.00 | 35 | ED | HED | 3 × 2 AGQ | 5 |
| Øvretveit et al. [69] | 12 | NO | 30.60 | 100 | Sport | Jiu-Jitsu | 3 × 2 AGQ-S | 7 |
| Rivera Pérez et al. [70] | 40 | ES | 10.87 | 48 | ED | PE | 3 × 2 AGQ-PE | 5 |
| Rivera Pérez et al. [71] | 1328 | ES | 13.11 | 51 | ED | PE | 3 × 2 AGQ-PE | 5 |
| Rivera Pérez et al. [72] | 1292 | ES | 13.05 | 51 | ED | PE | 3 × 2 AGQ-PE | 5 |
| Sari et al. [26] | 424 | ID | SEC | NR | ED | SEC | AGQ-R | 7 |
| Shen et al. [73] | 792 | CN | 16.93 | 46 | ED | PE | 3 × 2 AGQ-PE | 7 |
| Thomas [74] | 482 | US | 24.04 | 21 | ED | HED | 3 × 2 AGQ | 7 |
| Üztemur [75] | 259 | TR | SEC | 46 | ED | SEC | SS-O AGS | 5 |
| Van Yperen [76] | 647 | NL | 26.49 | 31 | Sport | Korfball | 3 × 2 AGQ-S | 7 |
| Wang et al. [77] | 475 | CN | 24.47 | 55 | Sport | HED | 3 × 2 AGQ-S | 7 |
| Wei et al. [78] | 406 | TW | 20.34 | 57 | Sport | HED | 3 × 2 AGQ-S (Chinese) | 7 |
| Wu [17] | 59 | TW | 5.00 | 49 | ED | PRI | PAGM | 4 |
| Yang and Cao [79] | 93 | US | HED | 25 | ED | HED | 3 × 2 AGQ | 7 |
| Yerdelen and Padir [20] | 207 | TR | NR | 45 | Profession | PRI/SEC | 3 × 2 AGQ-Teachers (Turkish) | 7 |
| Zhou et al. [25] | 177; 158; 348 | CN | 20.53, 20.64, 11.56 | 54, 38, 46 | ED | HED, PRI | 3 × 2 AGQ (Chinese), AGQ-short | 7 |

Abbreviations: AT = Austria, CN = China, DE = Germany, ES = Spain, FR = France, HU = Hungary, ID = Indonesia, KE = Kenya, NL = The Netherlands, NO = Norway, RU = Russian Federation, SA = Saudi Arabia, TR = Turkey, TW = Taiwan, US = United States of America, ZA = South Africa; ED = Education, HED = Higher Education, PRI = Primary, SEC = Secondary, AGQ = Achievement Goal Questionnaire, R= Revised, S = Sport, RS = Recreational Sport, PE = Physical Education, SS-O = Social Studies Oriented, PAGM = Pictorial Achievement Goal Measurement for Kindergarteners, CGSQ = Classroom Goal Structure Questionnaire, QTLG = Questionnaire on Teamwork Learning Goals.

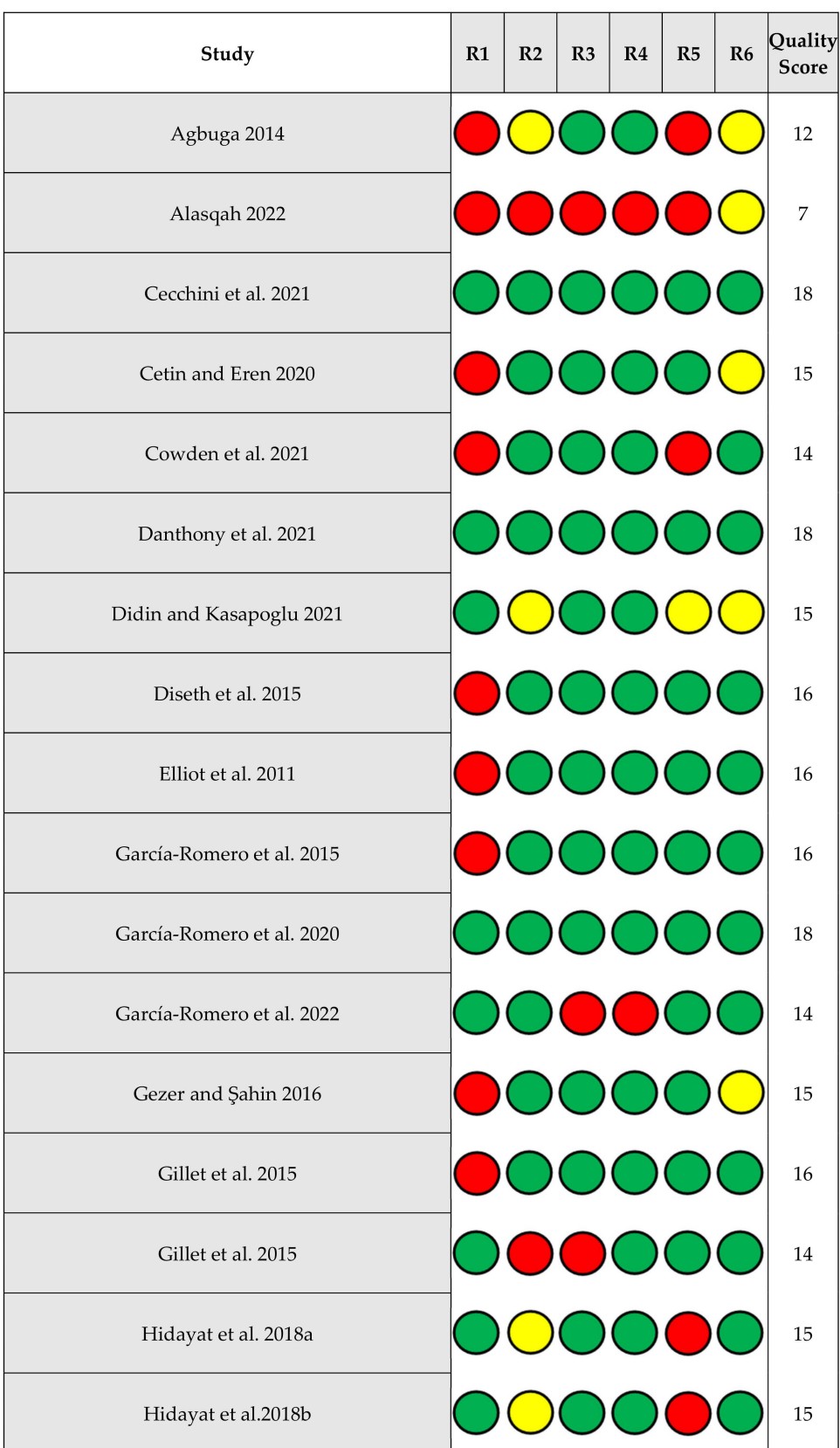

**Figure 3.** *Cont.*

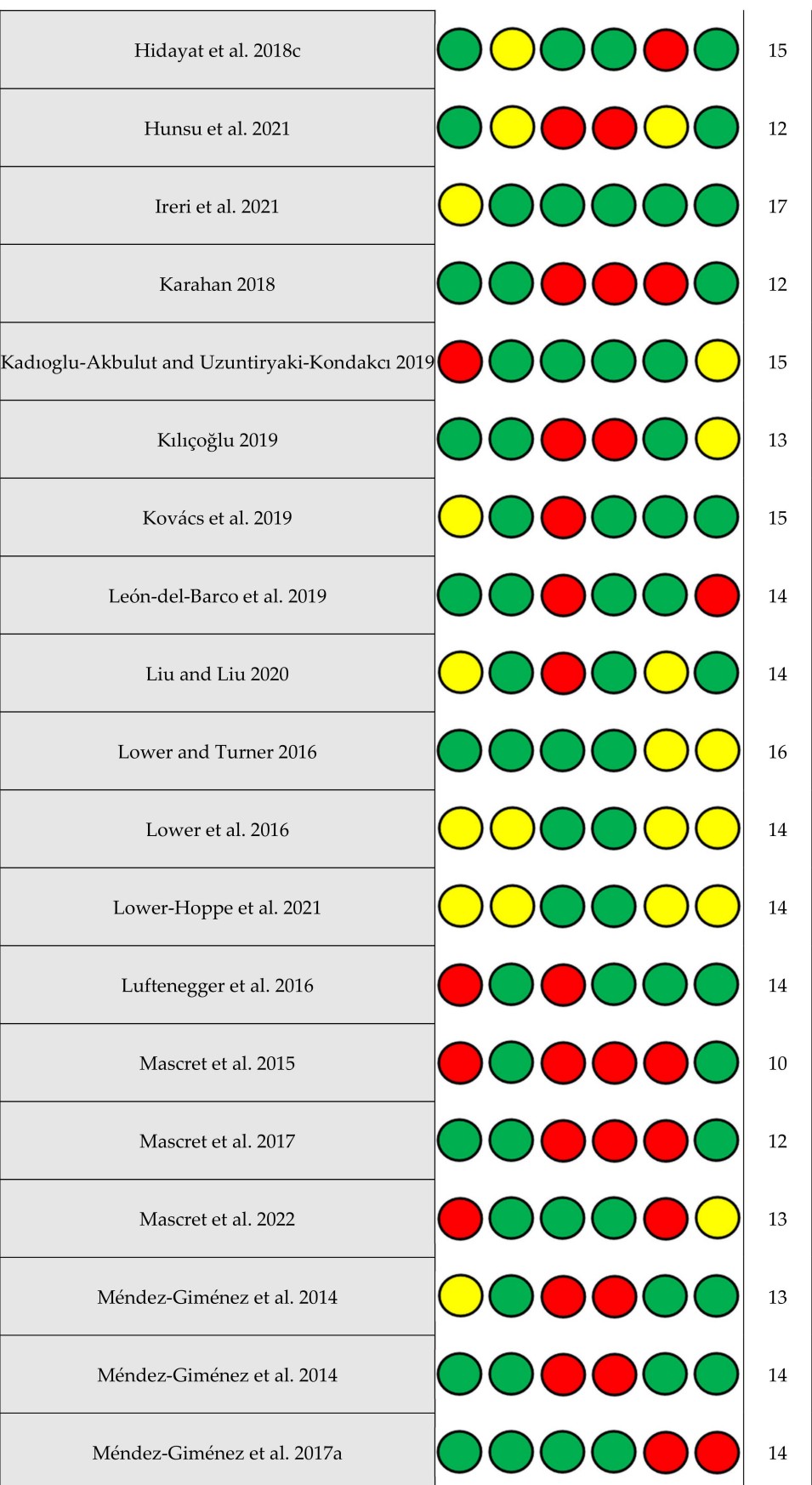

**Figure 3.** *Cont.*

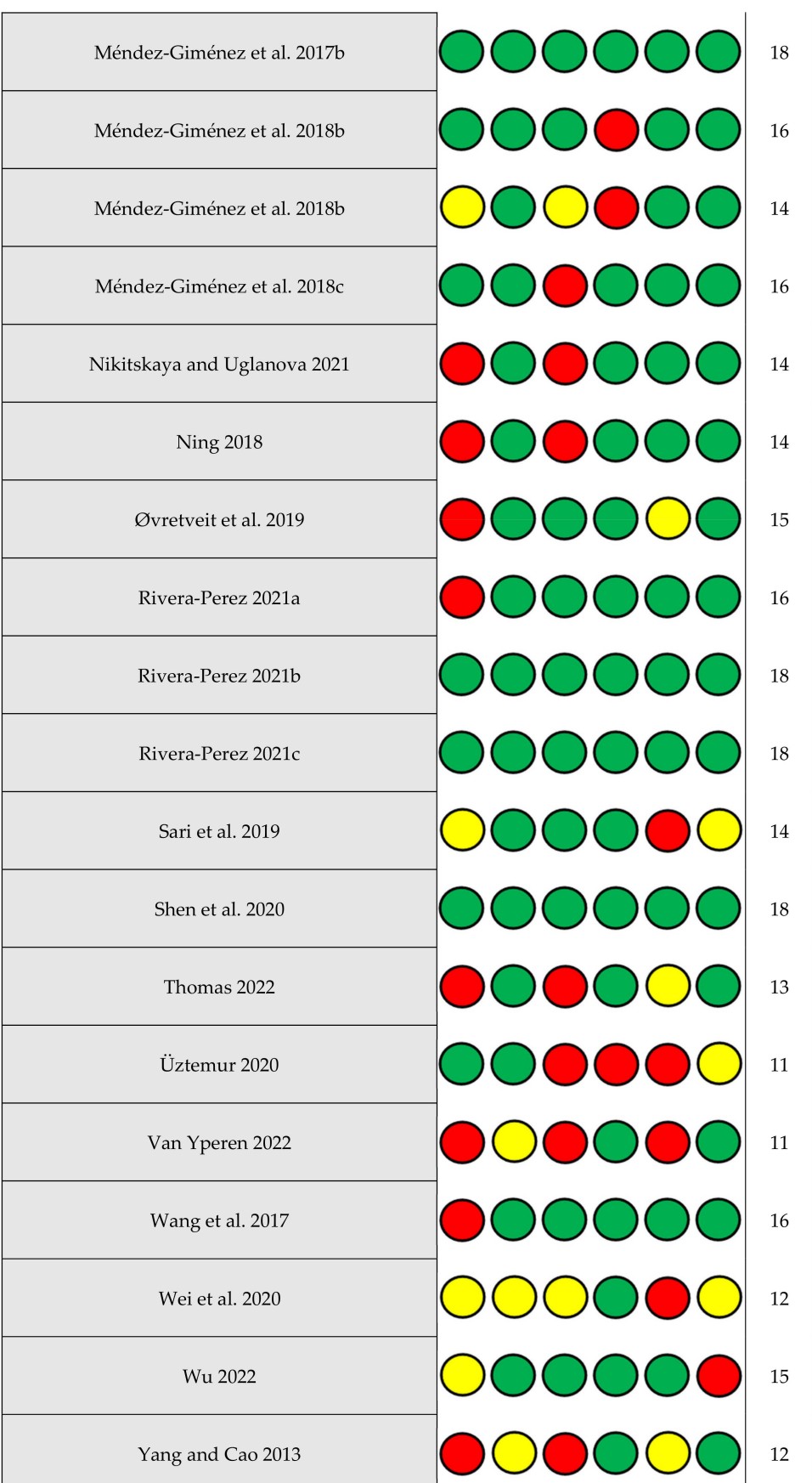

**Figure 3.** *Cont.*

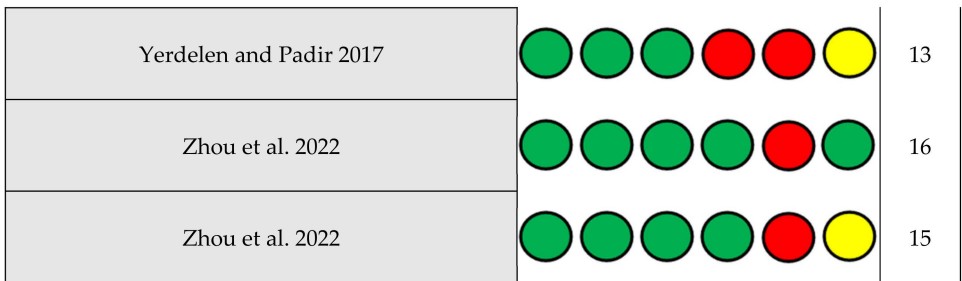

**Figure 3.** Risk of bias within studies [9,16–27,37–79]. Abbreviations: R1 = sample selection, R2 = nonresponse, R3 = collection, R4 = collection method, R5 = anonymity, R6 = AGQ measure; green circle = 3 points, yellow circle = 2 points, red circle = 1 point.

**Table 3.** Random-effects correlation (*r*) and [95% confidence intervals] data and averaged reliability coefficients [95% confidence intervals] across the diagonal for the 3 × 2 achievement goals across all studies contributing to the specific analysis.

| | | 1 | 2 | 3 | 4 | 5 | 6 |
|---|---|---|---|---|---|---|---|
| 1 | TAP | 0.82 [0.80, 0.84] | 0.59 [0.59, 0.60] | 0.66 [0.65, 0.67] | 0.50 [0.49, 0.51] | 0.36 [ 0.35, 0.37] | 0.34 [0.33, 0.36] |
| 2 | TAV | 0.59 [0.55, 0.62] | 0.81 [0.79, 0.83] | 0.54 [0.53, 0.55] | 0.66 [ 0.65, 0.67] | 0.27 [ 0.26, 0.28] | 0.47 [0.46, 0.48] |
| 3 | SAP | 0.62 [0.58, 0.66] | 0.52 [0.47, 0.55] | 0.83 [0.81, 0.85] | 0.61 [0.60, 0.61] | 0.36 [ 0.35, 0.37] | 0.37 [0.36, 0.38] |
| 4 | SAV | 0.47 [0.43, 0.51] | 0.64 [0.58, 0.70] | 0.58 [0.54, 0.62] | 0.81 [0.78, 0.83] | 0.35 [0.34, 0.36] | 0.56 [0.55, 0.57] |
| 5 | OAP | 0.34 [0.29, 0.39] | 0.27 [0.21, 0.33] | 0.35 [0.30, 0.40] | 0.35 [0.29, 0.41] | 0.87 [0.86, 0.89] | 0.67 [0.67, 0.68] |
| 6 | OAV | 0.32 [0.27, 0.36] | 0.44 [0.37, 0.49] | 0.34 [0.30, 0.38] | 0.54 [0.49, 0.59] | 0.65 [0.61, 0.69] | 0.85 [0.84, 0.87] |

Abbreviations: TAP = task approach, TAV = task avoidance, SAP = self-approach, SAV = self-avoidance, OAP = other approach, OAV = other avoidance. Notes: Fixed effects on top half of matrix, random effects on bottom half of matrix, Cronbach coefficients on the diagonal. Studies contributing at least one Cronbach coefficient [9,19–22,25–27,30,37–43,46,47,50–52,54–59,61,63–68,70–74,76–79]. Studies contributing at least one intercorrelation [9,19–22,25–27,39,40,42,44–47,49,51,52,54–56,58,59,61–67,69,71,73–76,78,79].

**Table 4.** Summary of mean fixed- and random-effects and publication bias statistics for each 3 × 2 achievement goal subscale scored on a 7-point Likert scale.

| | | | Mean Statistics | | | | Publication Bias Statistics | | |
|---|---|---|---|---|---|---|---|---|---|
| Goal | Model | *k* | *M* [95% CI] | *SE* | Z-Value | $I^2$ | Fail–Safe *n* | Trim *n* | *M* [95% CI] |
| Task approach | F | 34 | 5.67 [5.65, 5.68] | 0.01 | 821.20 | 99.85 | | | |
| | R | 34 | 5.74 [5.38, 6.09] | 0.18 | 31.48 | | >1 k | 5 L | 5.58 [5.24, 5.91] |
| Task avoidance | F | 33 | 5.35 [5.33, 5.37] | 0.01 | 665.97 | 99.80 | | | |
| | R | 33 | 5.55 [5.18, 5.92] | 0.19 | 29.64 | | >1 k | 5 L | 5.37 [5.03, 5.72] |
| Self-approach | F | 33 | 5.49 [5.48, 5.51] | 0.01 | 689.91 | 99.78 | | | |
| | R | 33 | 5.48 [5.13, 5.82] | 0.18 | 31.00 | | >1 k | 8 L | 5.19 [4.84, 5.55] |
| Self-avoidance | F | 32 | 5.30 [5.29, 5.32] | 0.01 | 584.59 | 99.76 | | | |
| | R | 32 | 5.26 [4.88, 5.64] | 0.19 | 27.29 | | >1 k | 6 L | 5.03 [4.63, 5.43] |
| Other approach | F | 34 | 4.13 [4.11, 4.15] | 0.01 | 406.51 | 99.71 | | | |
| | R | 34 | 4.53 [4.14, 4.92] | 0.20 | 22.83 | | >1 k | 5 L | 4.31 [3.93, 4.68] |
| Other avoidance | F | 33 | 4.36 [4.34, 4.38] | 0.01 | 443.68 | 99.65 | | | |
| | R | 33 | 4.58 [4.24, 4.93] | 0.17 | 26.21 | | >1 k | 1 L | 4.52 [4.16, 4.88] |

Note: All Z-value *p* < 0.001; Abbreviations: F = Fixed, R = Random, *k* = number of samples, *M* = mean, CI = confidence interval, *SE* = standard error, $I^2$ = heterogeneity statistic.

**Table 5.** Summary of mean fixed- and random-effects and publication bias statistics for each 3 × 2 achievement goal subscale scored on a 5-point Likert scale.

| Goal | Model | k | M [95% CI] | SE | Z-Value | I² | Fail–Safe n | Trim n | M [95% CI] |
|------|-------|---|-----------|----|---------|----|------------|--------|-----------|
| | | | **Mean Statistics** | | | | **Publication Bias Statistics** | | |
| Task approach | F | 21 | 4.07 [4.05, 4.08] | 0.01 | 534.19 | 98.34 | | | |
| | R | 21 | 4.11 [3.99, 4.23] | 0.06 | 66.34 | | >1 k | 4 L | 4.04 [3.92, 4.16] |
| Task avoidance | F | 19 | 3.96 [3.95, 3.98] | 0.01 | 432.25 | 93.71 | | | |
| | R | 19 | 3.99 [3.91, 4.07] | 0.04 | 100.90 | | >1 k | 1 L | 3.99 [3.91, 4.07] |
| Self-approach | F | 21 | 3.99 [3.97, 4.00] | 0.01 | 513.07 | 95.76 | | | |
| | R | 21 | 4.01 [3.93, 4.09] | 0.04 | 99.15 | | >1 k | 4 L | 3.97 [3.89, 4.05] |
| Self-avoidance | F | 19 | 3.71 [3.69, 3.72] | 0.01 | 377.69 | 96.00 | | | |
| | R | 19 | 3.69 [3.58, 3.79] | 0.05 | 70.75 | | >1 k | 5 L | 3.60 [3.50, 3.70] |
| Other approach | F | 19 | 3.21 [3.19, 3.23] | 0.01 | 286.00 | 98.00 | | | |
| | R | 19 | 3.30 [3.14, 3.46] | 0.08 | 40.08 | | >1 k | 0 | 3.30 [3.14, 3.46] |
| Other avoidance | F | 19 | 3.43 [3.40, 3.45] | 0.01 | 315.17 | 94.45 | | | |
| | R | 19 | 3.48 [3.39, 3.58] | 0.05 | 71.59 | | >1 k | 0 | 3.48 [3.39, 3.58] |

Note: All Z-value $p < 0.001$; Abbreviations: F = Fixed, R = Random, $k$ = number of samples, $M$ = mean, CI = confidence interval, $SE$ = standard error, 95% LL = confidence interval lower limit, 95% UL = 95% confidence interval upper limit, $I^2$ = heterogeneity statistic.

### 3.4. Hypothesis 2 Results

Our second hypothesis concerned our subgroup analyses based on domain and the compulsory nature of the participant sub-domain. Sufficient samples only existed for the 7-point response scale datasets. To assess for statistical differences for goal valence and for each individual achievement goal, we conducted group mixed-effect analyses (see Table 6). Significant and medium-to-large effect size differences resulted in sport being higher compared to education in all approach goals combined (*g* large), the self-approach goal (*g* medium), and other approach goal (*g* large).

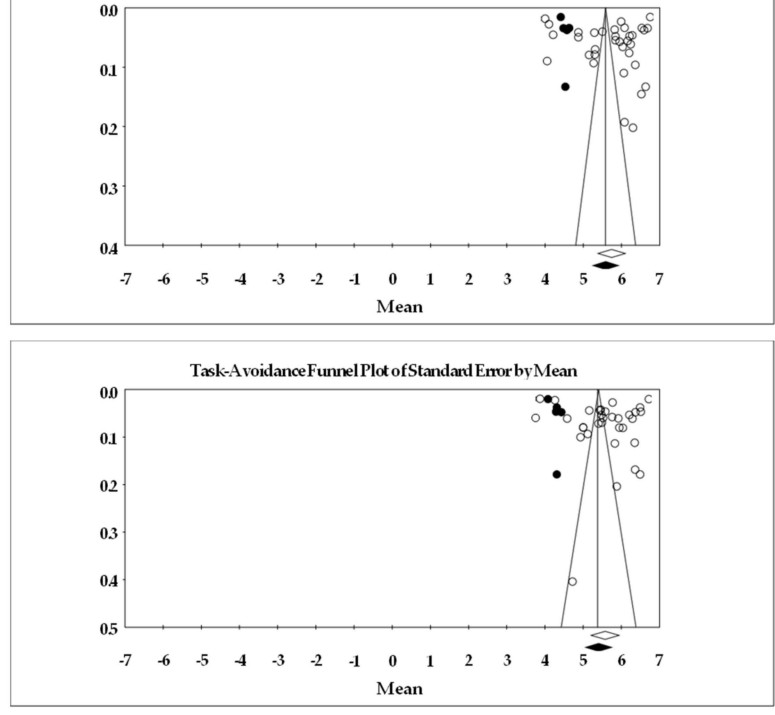

**Figure 4.** *Cont.*

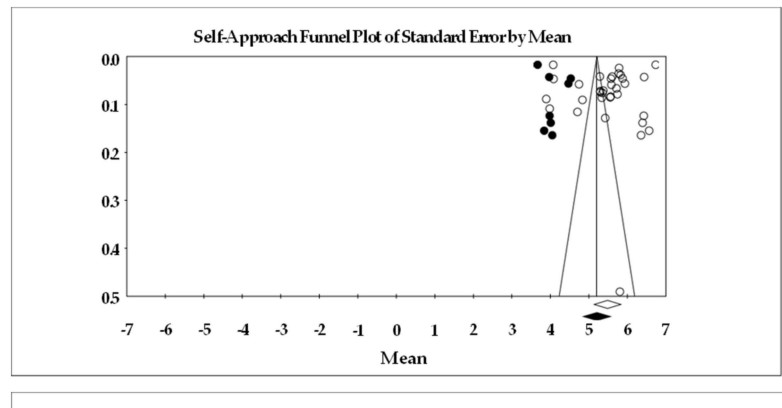

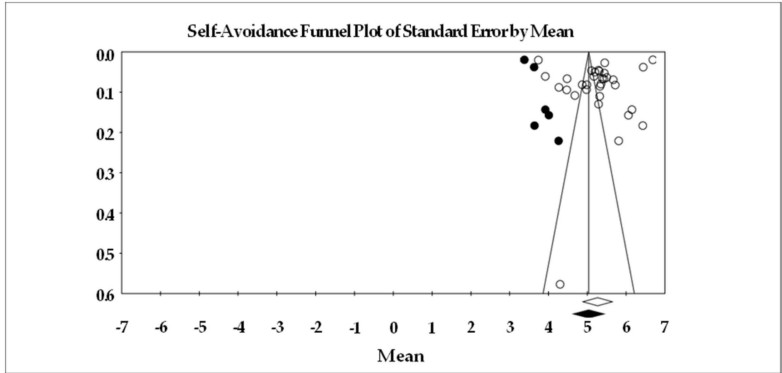

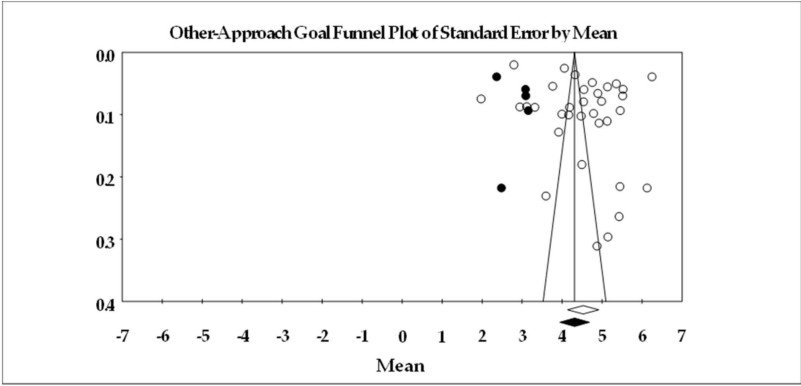

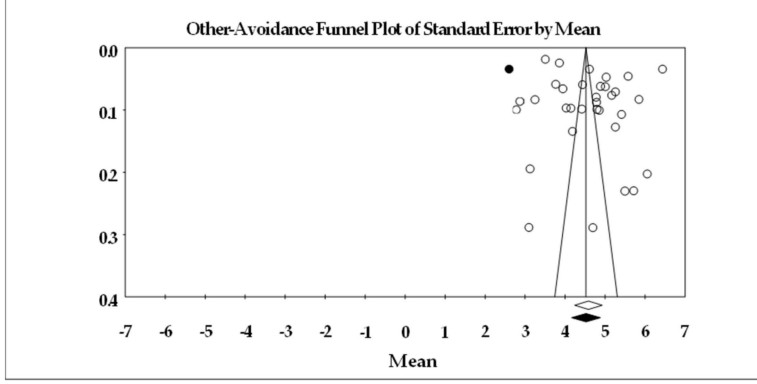

**Figure 4.** Random-effects funnel plot of standard error by means for each 3 × 2 achievement goal scored on a 7-point scale. Clear circles are the observed data, filled in circles are the imputed data. Articles contributing to each as follows: task approach [9,19–21,25–27,40,42,43,49,51,54,55,57,58,61,64, 67,69,73,74,76–79], task avoidance [9,19–21,25–27,40,42,43,49,51,54,55,57,58,61,64,67,69,73,74,76–78], self-approach [9,19–21,25–27,40,42,43,49,51,54,55,57,58,61,64,67,69,73,74,76–79], self-avoidance [9,19–21,25–27,40,42,43,49,51,54,55,57,58,61,64,67,69,73,74,76–78], other approach [9,19–21,25–27,40,42,43, 49,51,54,55,57,58,61,64,67,69,73,74,76–79], other avoidance [9,19–21,25–27,40,42,43,49,51,54,55,57,58, 61,64,67,69,73,74,76–78].

**Table 6.** Education compared with sport participant group summary of mixed-effects statistics for the approach, avoidance, and each 3 × 2 achievement goal subscale.

| Achievement Goal | Group | k | M | SE | 95% LL | 95% UL | Q$_{TB}$ | *p*-Value | Hedges' *g* |
|---|---|---|---|---|---|---|---|---|---|
| Approach Goals | Education | 22 | 4.91 | 0.23 | 4.45 | 5.37 | | | |
| | Sport | 11 | 5.67 | 0.09 | 5.49 | 5.84 | 8.98 | 0.00 | 0.84 |
| Avoidance Goals | Education | 21 | 4.91 | 0.24 | 4.44 | 5.37 | | | |
| | Sport | 11 | 5.16 | 0.13 | 4.90 | 5.42 | 0.85 | 0.36 | 0.27 |
| Task approach | Education | 20 | 5.56 | 0.26 | 5.04 | 6.07 | | | |
| | Sport | 10 | 6.02 | 0.11 | 5.81 | 6.23 | 2.62 | 0.11 | 0.47 |
| Task avoidance | Education | 19 | 5.43 | 0.26 | 4.91 | 5.95 | | | |
| | Sport | 10 | 5.64 | 0.15 | 5.34 | 5.94 | 0.47 | 0.49 | 0.22 |
| Self-approach | Education | 19 | 5.28 | 0.26 | 4.77 | 5.79 | | | |
| | Sport | 10 | 5.89 | 0.15 | 5.59 | 6.19 | 4.15 | 0.04 | 0.63 |
| Self-avoidance | Education | 18 | 5.16 | 0.28 | 4.60 | 5.71 | | | |
| | Sport | 10 | 5.50 | 0.15 | 5.21 | 5.79 | 1.16 | 0.28 | 0.34 |
| Other approach | Education | 19 | 4.31 | 0.26 | 3.80 | 4.82 | | | |
| | Sport | 11 | 5.14 | 0.15 | 4.85 | 5.44 | 7.75 | 0.01 | 0.87 |
| Other avoidance | Education | 18 | 4.53 | 0.25 | 4.04 | 5.01 | | | |
| | Sport | 11 | 4.71 | 0.19 | 4.34 | 5.09 | 0.35 | 0.55 | 0.19 |

Abbreviations: *k* = number of samples, *M* = mean, *SE* = standard error, 95% LL = 95% confidence interval lower limit, 95% UL = 95% confidence interval upper limit, Q$_{TB}$ = Q total between.

The second part of this hypothesis concerned our subgroup analyses based on the compulsory nature of the participant sub-domain. As with the sport compared to education datasets, sufficient samples only existed for the 7-point response scale datasets. To evaluate statistical differences for goal valence and for each individual achievement goal, we conducted group mixed-effect analyses (see Table 6). Many significant and large to-very-large effect size differences resulted in non-compulsory being higher compared to compulsory in all approach goals combined (*g* very large), all avoidance goals (*g* large), the task approach (*g* very large) and avoidance goal (*g* large), and the other approach goal (*g* large). Examining the Hedges' *g* values in Table 7, though not significant, the differences between the two groups, non-compulsory and compulsory, were medium in meaningfulness.

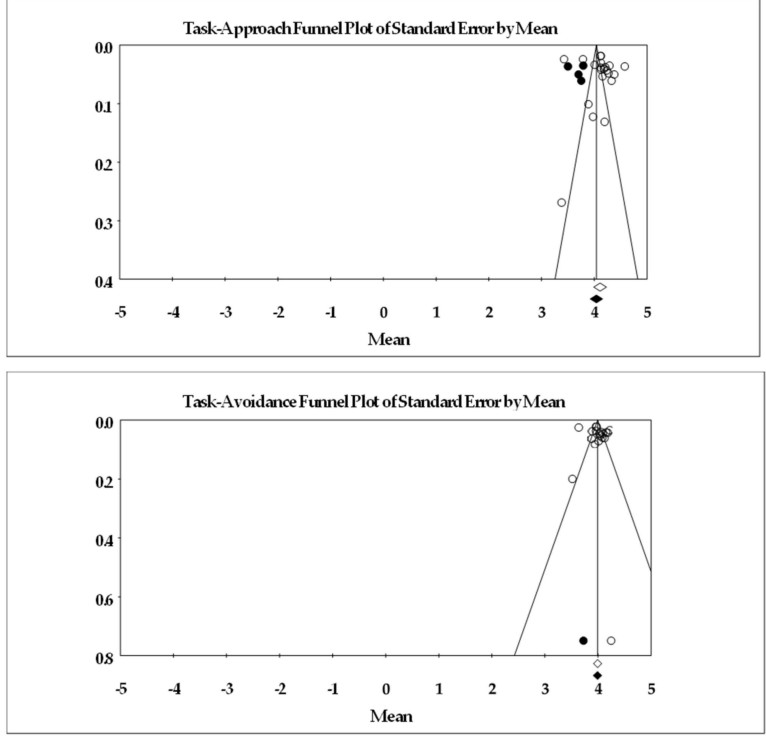

**Figure 5.** *Cont.*

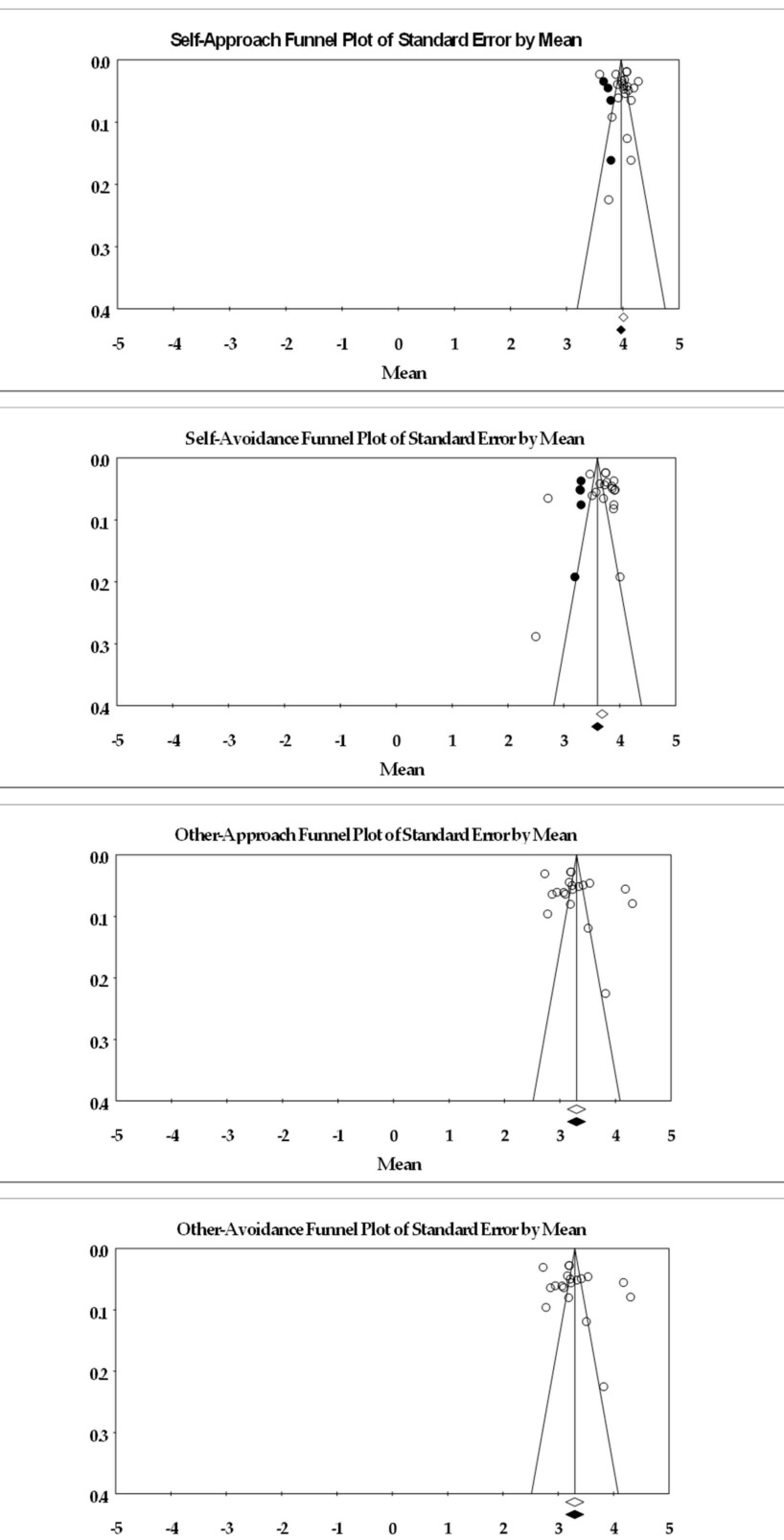

**Figure 5.** Random-effects funnel plot of standard error by means for each 3 × 2 achievement goal scored on a 5-point scale. Clear circles are the observed data, filled in circles are the imputed data. Articles contributing to each as follows: task approach [22,38,39,44,46,47,52,53,56,63,65,66,70,71,75], task avoidance [22,38,39,44,46,47,52,53,56,63,65,66,75], self-approach [22,38,39,44,46,47,52,53,56,63,65, 66,70,71,75], self-avoidance [22,38,39,44,46,47,52,53,56,63,65,66,75], other approach [22,38,39,44,46,47, 52,53,56,63,65,66,75], other avoidance [22,38,39,44,46,47,52,53,56,63,65,66,75].

**Table 7.** Compulsory (no, yes) mixed-effects statistics for the approach, avoidance, and each 3 × 2 achievement goal subscale.

| Achievement Goal | Group | k | M | SE | 95% LL | 95% UL | Q_TB | p-Value | Hedges' g |
|---|---|---|---|---|---|---|---|---|---|
| Approach Goals | No | 25.00 | 5.43 | 0.13 | 5.18 | 5.67 | | | |
| | Yes | 10.00 | 4.44 | 0.28 | 3.88 | 4.99 | 10.15 | 0.00 | 1.37 |
| Avoidance Goals | No | 24.00 | 5.23 | 0.16 | 4.92 | 5.53 | | | |
| | Yes | 10.00 | 4.47 | 0.26 | 3.96 | 4.99 | 6.18 | 0.01 | 0.96 |
| Task approach | No | 24.00 | 5.92 | 0.14 | 5.65 | 6.18 | | | |
| | Yes | 8.00 | 5.02 | 0.40 | 4.23 | 5.80 | 4.56 | 0.03 | 1.29 |
| Task avoidance | No | 23.00 | 5.72 | 0.14 | 5.44 | 5.99 | | | |
| | Yes | 8.00 | 4.94 | 0.37 | 4.22 | 5.65 | 3.93 | 0.05 | 1.00 |
| Self-approach | No | 24.00 | 5.58 | 0.15 | 5.29 | 5.88 | | | |
| | Yes | 7.00 | 5.08 | 0.41 | 4.28 | 5.88 | 1.34 | 0.25 | 0.61 |
| Self-avoidance | No | 23.00 | 5.39 | 0.17 | 5.05 | 5.73 | | | |
| | Yes | 7.00 | 4.94 | 0.44 | 4.07 | 5.80 | 0.91 | 0.34 | 0.50 |
| Other approach | No | 24.00 | 4.81 | 0.17 | 4.48 | 5.14 | | | |
| | Yes | 8.00 | 3.95 | 0.31 | 3.34 | 4.56 | 5.89 | 0.02 | 1.02 |
| Other avoidance | No | 23.00 | 4.75 | 0.20 | 4.36 | 5.15 | | | |
| | Yes | 8.00 | 4.24 | 0.24 | 3.78 | 4.70 | 2.77 | 0.10 | 0.57 |

Abbreviations: $k$ = number of samples, $M$ = mean, $SE$ = standard error, 95% LL = 95% confidence interval lower limit, 95% UL = 95% confidence interval upper limit, $Q_{TB}$ = Q total between.

### 3.5. Hypothesis 3 Results

For our last hypothesis, we examined the relationships between the achievement goals and correlates found in the literature. We formed five categories based on the individual correlation measures such as individual differences, desired and undesired motivations, facilitative and debilitative learning strategies. From the initial list and now knowing the number of samples per correlated, we formed the following categories: facilitative learning strategies (e.g., effort, metacognition), desired motivations (e.g., intrinsic motivation, competence needs), positive (satisfaction, self-determination index) and negative emotions (e.g., test anxiety, worry), and performance (academic, sport). Our aim was 10 samples per correlate analysis. For performance, we did not meet this threshold. For our other correlates such as individual differences and undesired motivations, we did not reach five samples and hence did not analyze these data. Our Supplementary Materials contain all of the extracted correlates.

Figure 6 contains the forest plots of all studies contributing to a correlate analysis for the combination of all approach and all avoidance goals across the following correlate categories: facilitative learning strategies, desired motivations, positive and negative emotions, and performance. Table 8 contains the effect size statistics and publication bias statistics for all of the analyzed correlates. The corresponding funnel plots are located in our Supplementary Materials. The majority of the meta-analyzed correlations were significant with minimal bias based on the fail–safe $n$, Orwin's $n$ and the trim $n$ values. Only the approach goals and facilitative learning strategies and desired motivation meta-analyzed correlations were medium in meaningfulness (i.e., >0.30). The approach goal and negative emotions and avoidance goals and performance were the least reliable (i.e., confidence intervals crossed 0). The approach goals compared were most related to the correlates. Minimal differences emerged between the approach and avoidance goals with positive and negative emotions.

Figures contain the forest plots of all studies contributing to a correlate analysis for the combination for each goal across the following correlate categories: facilitative learning strategies (Figure 7), desired motivations (Figure 8), positive (Figure 9) and negative (Figure 10) emotions, and performance (Figure 11). Table 9 contains the effect size statistics and publication bias statistics for all of the analyzed correlates. The corresponding funnel plots are located in our Supplementary Materials. Across all 30 computed fixed-effect values, the majority ($n$ = 17) were small in meaningfulness followed by negligible ($n$ = 9) and medium ($n$ = 4) in meaningfulness. The larger (i.e., medium-sized correlations) results were observed with task and self-approach goals with facilitative learning strate-

gies and desired motivations. The categorization changed for the self-approach goal and performance, with meaningfulness dropping from small to negligible. The trim and filled mean statistics suggested one more medium in meaningfulness value (i.e., task avoidance goal and positive emotions) and two more small in meaningfulness values, from negligible, for again the task avoidance goal, this time with performance and the other avoidance goal and negative emotions.

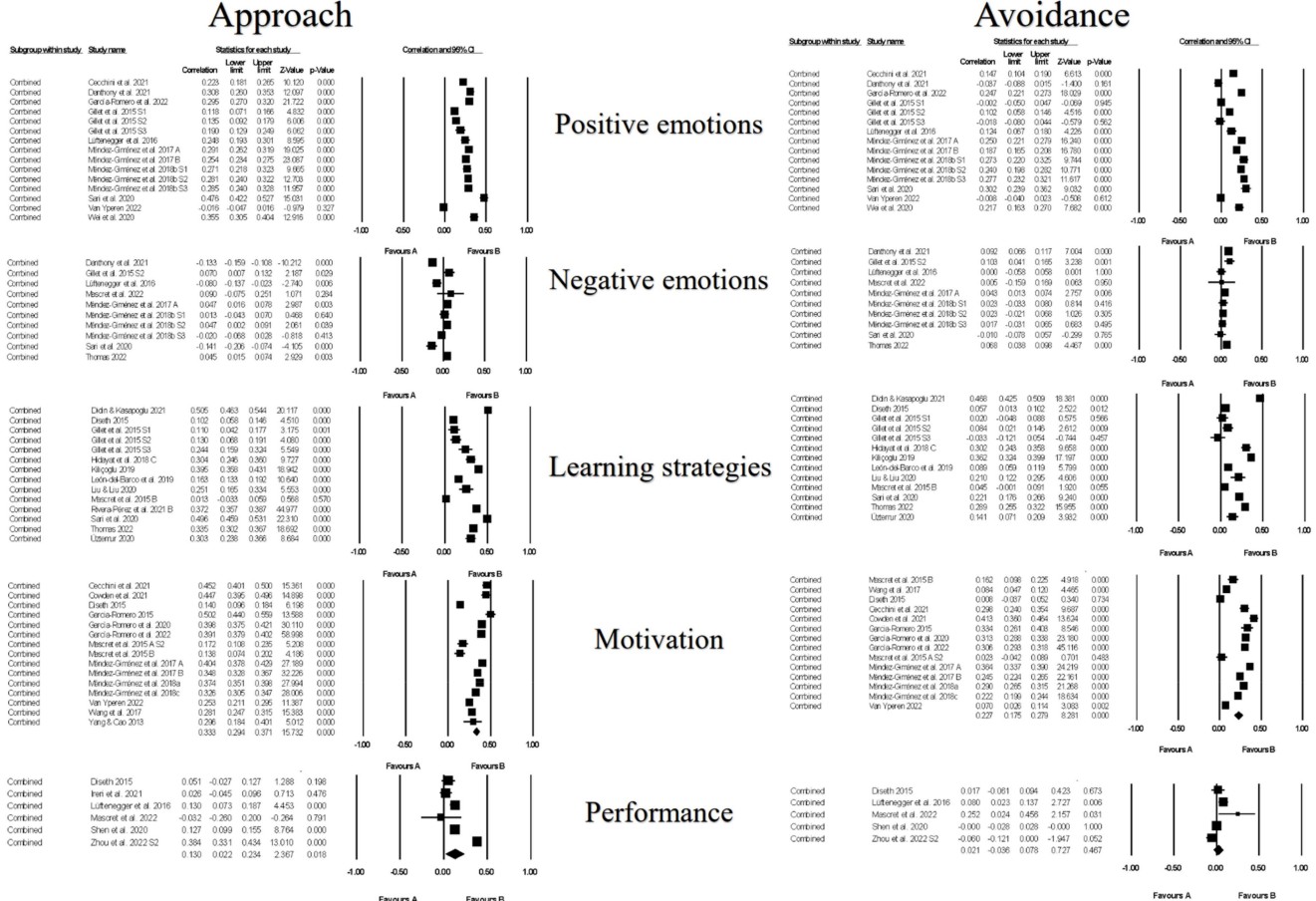

**Figure 6.** Random-effects individual correlates for the 3 × 2 achievement goals collapsed into approach and avoidance goals by the positive emotions, negative emotions, learning strategies, motivation, and performance. Studies contributing to each correlate analyses are as follows: positive emotions approach and avoidance [26,27,39,43,48,61,63,64,66,76,78], negative emotions approach and avoidance [26,27,43,61–63,66,74], learning strategies approach [18,19,26,27,44,45,51,56,58,71,74, 75], learning strategies avoidance [18,19,26,27,44,45,51,56,58,74,75], motivation approach [19,21,39, 42,45–48,63–65,67,76,77,79], motivation avoidance [19,21,39,42,45–48,63–65,67,76,77], performance approach [25,45,53,61,62,73], and performance avoidance [25,45,61,62,73].

**Table 8.** Effects size and publication bias statistics for the correlates with the approach and avoidance subscales merged.

| Goal | Model | k | r [95% CI] | Z | I² | Fail–Safe n | Orwin's n | Trim n | Mean [95% CI] |
|---|---|---|---|---|---|---|---|---|---|
| | | | **Effect Size Statistics** | | | **Publication Bias Statistics** | | | |
| | | | *Facilitative Learning Strategies* | | | | | | |
| Approach | Fixed | 14 | 0.31 [0.30, 0.32] | 57.83 | 98.09 | | | | |
| | Random | 14 | 0.27 [0.19, 0.35] | 6.54 | | 1635 | 31 | 0 | |
| Avoidance | Fixed | 13 | 0.19 [0.17, 0.20] | 26.91 | 97.22 | | | | |
| | Random | 13 | 0.18 [0.10, 0.26] | 4.20 | | 3092 | 12 | 1 L | 0.19 [0.11, 0.27] |
| | | | *Desired Motivations* | | | | | | |
| Approach | Fixed | 15 | 0.36 [0.35, 0.37] | 92.22 | 95.94 | | | | |
| | Random | 15 | 0.33 [0.29, 0.37] | 15.73 | | 32,449 | 41 | 0 | |
| Avoidance | Fixed | 14 | 0.26 [0.25, 0.27] | 65.75 | 97.56 | | | | |
| | Random | 14 | 0.23 [0.17, 0.28] | 8.28 | | 4762 | 24 | 0 | |
| | | | *Positive Emotions* | | | | | | |
| Approach | Fixed | 15 | 0.17 [0.16, 0.18] | 32.04 | 96.63 | | | | |
| | Random | 15 | 0.25 [0.19, 0.30] | 8.46 | | 1027 | 21 | 3 L | 0.21 [0.16, 0.27] |
| Avoidance | Fixed | 15 | 0.17 [0.16, 0.18] | 32.04 | 96.63 | | | | |
| | Random | 15 | 0.16 [0.10, 0.21] | 5.38 | | 4680 | 10 | 0 | |
| | | | *Negative Emotions* | | | | | | |
| Approach | Fixed | 10 | −0.02 [−0.03, −0.01] | −3.05 | 93.91 | | | | |
| | Random | 10 | −0.01 [−0.07, 0.05] | −0.35 | | 5 | | 0 | |
| Avoidance | Fixed | 10 | −0.02 [−0.03, −0.01] | −3.05 | 93.91 | | | | |
| | Random | 10 | −0.01 [−0.07, 0.05] | −0.35 | | 135 | | 2 R | 0.05 [0.03, 0.08] |
| | | | *Performance* | | | | | | |
| Approach | Fixed | 6 | 0.15 [0.12, 0.16] | 13.34 | 94.44 | | | | |
| | Random | 6 | 0.13 [0.02, 0.23] | 2.37 | | 284 | 3 | 2 R | 0.18 [0.08, 0.27] |
| Avoidance | Fixed | 5 | 0.01 [−0.01, 0.03] | 0.67 | 74.37 | | | | |
| | Random | 5 | 0.02 [−0.04, 0.08] | 0.73 | | 0 | | 0 | |

Note: *k* = number of samples, *SE* = standard error, CI = confidence interval, *I²* = heterogeneity statistic.

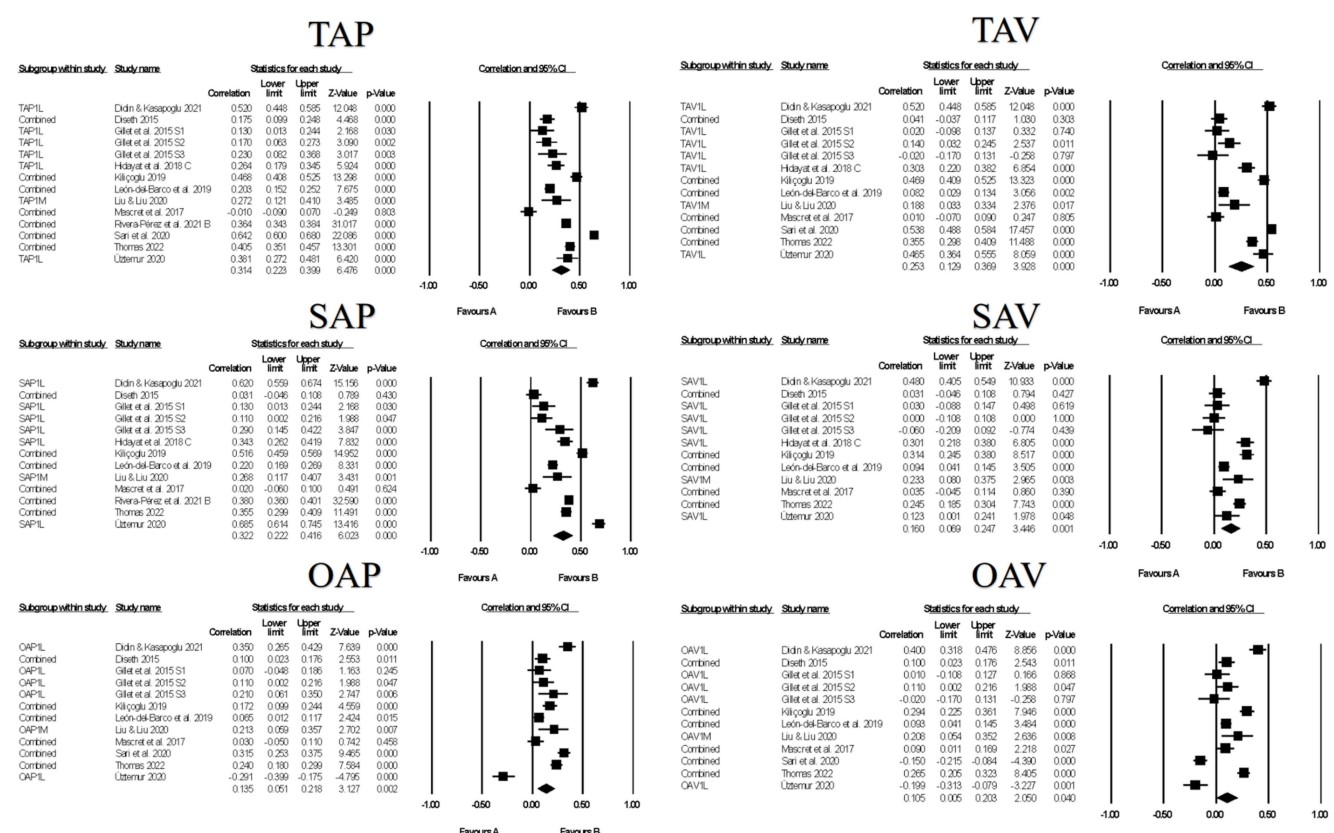

**Figure 7.** Random-effects individual correlates for the 3 × 2 achievement goals and facilitative learning strategies. Studies contributing to each correlate analyses are as follows: Task approach—TAP [18,19,26,27,44,45,51,56,58,71,74,75], Task avoidance—TAV [18,19,26,27,44,45,51,56,58,74,75], Self-approach—SAP [18,19,27,44,45,51,56,58,71,74,75], Self-avoidance—SAV [18,19,27,44,45,51,56,58,74,75], Other approach—OAP [18,19,26,27,44,45,56,58,74,75], Other avoidance—OAV [18,19,26,27,44,45,56,58,74,75].

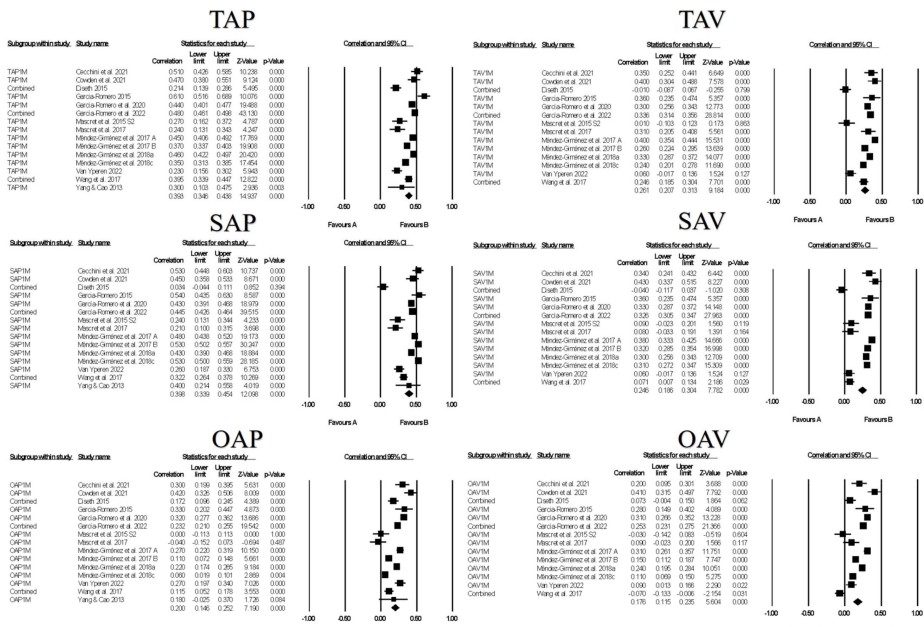

**Figure 8.** Random-effects individual correlates for the 3 × 2 achievement goals and desired motivations. Studies contributing to each correlate analyses are as follows: Task approach—TAP [19,21,39, 42,45–48,63–65,67,76,77,79], Task avoidance—TAV [19,21,39,42,45–48,63–65,67,76,77], Self-approach—SAP [19,21,39,42,45–48,63–65,67,76,77,79], Self-avoidance—SAV [19,21,39,42,45–48,63–65,67,76,77], Other approach—OAP [19,21,39,42,45–48,63–65,67,76,77,79], Other avoidance—OAV [19,21,39,42,45–48,63–65,67,76,77].

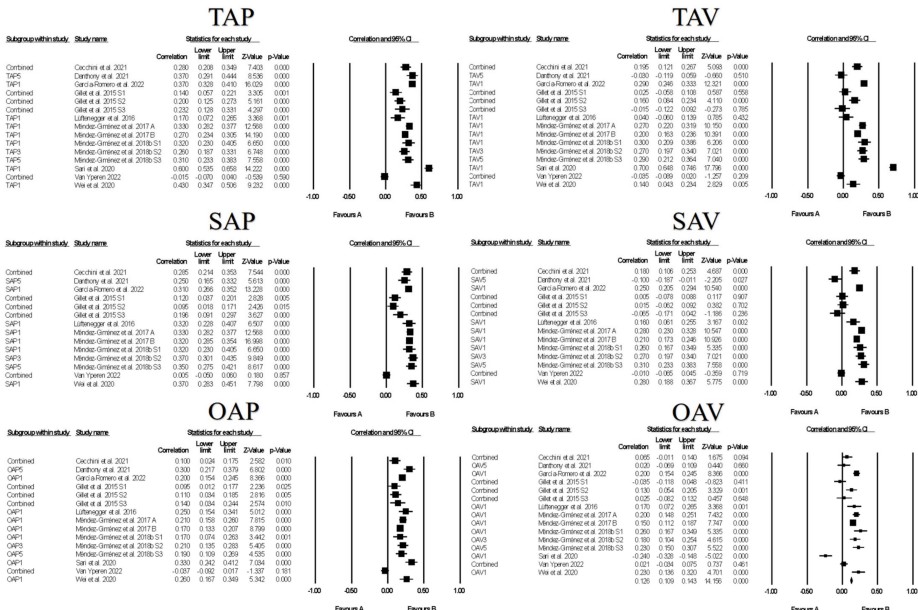

**Figure 9.** Random-effects individual correlates for the 3 × 2 achievement goals and positive emotions. Studies contributing to each correlate analyses are as follows: Task approach—TAP [26,27,39,43,48,61,63,64,66,76,78], Task avoidance—TAV [26,27,39,43,48,61,63,64,66,76,78], Self-approach—SAP [27,39,43,48,61,63,64,66,76,78], Self-avoidance—SAV [27,39,43,48,61,63,64,66,76,78], Other approach—OAP [26,27,39,43,48,61,63,64,66,76,78], Other avoidance—OAV [26,27,39,43,48,61, 63,64,66,76,78].

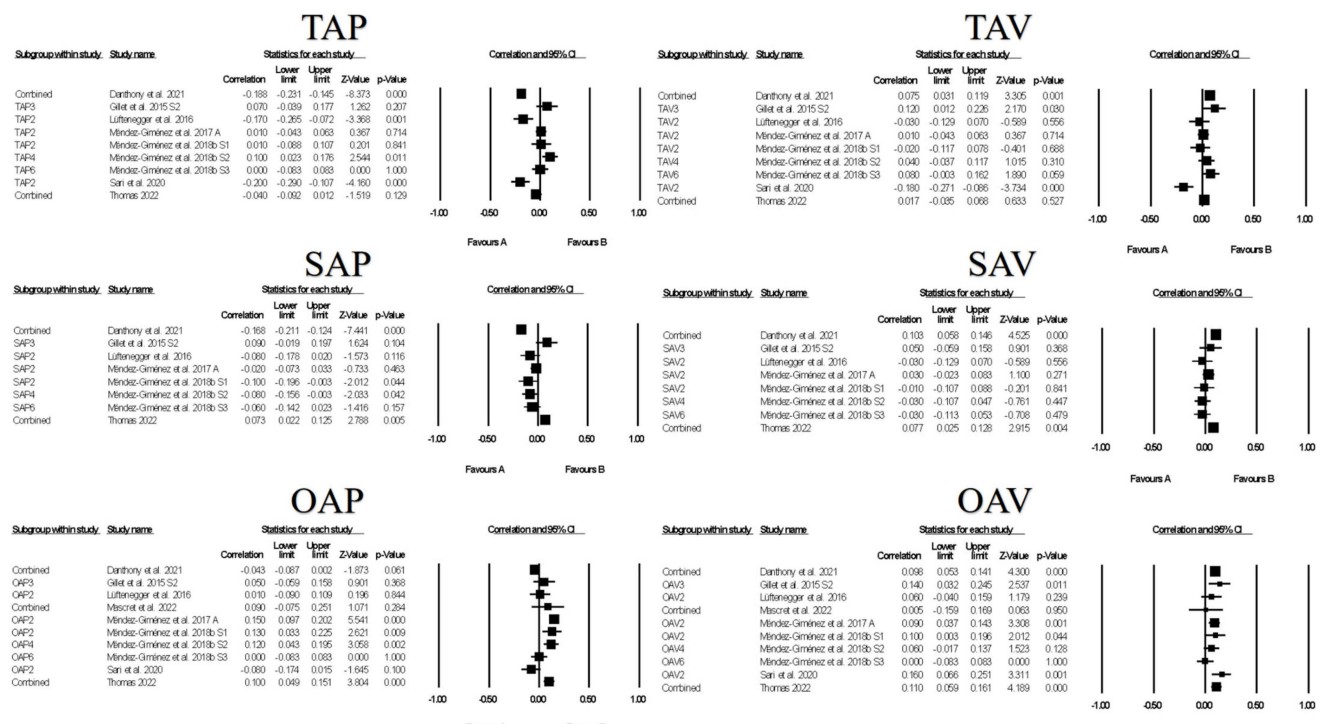

**Figure 10.** Random-effects individual correlates for the 3 × 2 achievement goals and negative emotions. Studies contributing to each correlate analyses are as follows: Task approach—TAP [26,27,43, 61,63,66,74], Task avoidance—TAV [26,27,43,61,63,66,74], Self-approach—SAP [27,43,61,63,66], Self-avoidance—SAV [27,43,61,63,66], Other approach—OAP [26,27,43,61–63,66,74], Other avoidance—OAV [26,27,43,61–63,66,74].

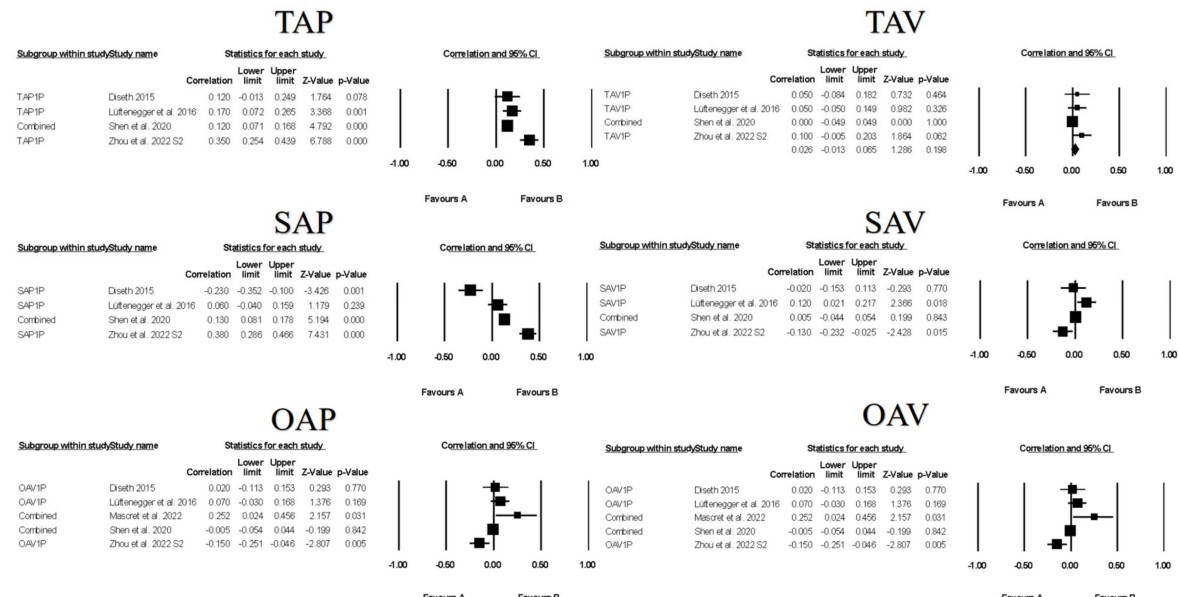

**Figure 11.** Random-effects individual correlates for the 3 × 2 achievement goals and performance. Studies contributing to each correlate analyses are as follows: Studies contributing to each correlate analyses are as follows: Task approach—TAP [25,45,61,73], Task avoidance—TAV [25,45,61,73], Self-approach—SAP [25,45,61,73], Self-avoidance—SAV [25,45,61,73], Other approach—OAP [25,45,61,62,73], Other avoidance—OAV [25,45,61,62,73].

**Table 9.** Effects size and publication bias statistics for correlates with each of the 3 × 2 subscales.

| Achievement Goal | Model | k | r [95% CI] | Z | I² | Fail–safe n | Orwin's n | Trim n | Mean [95% CI] |
|---|---|---|---|---|---|---|---|---|---|
| | | | | | Effect Size Statistics | Publication Bias Statistics | | | |
| | | | | Facilitative Learning Strategies | | | | | |
| TAP | Fixed | 14.00 | 0.34 [0.33, 0.36] | 42.30 | 96.28 | | | | |
| | Random | 14.00 | 0.31 [0.22, 0.40] | 6.48 | | 6018 | 37 | 2 R | 0.35 [0.26, 0.44] |
| TAV | Fixed | 13.00 | 0.27 [0.24, 0.29] | 23.14 | 96.65 | | | | |
| | Random | 13.00 | 0.25 [0.13, 0.37] | 3.93 | | 2268 | 23 | 0 | |
| SAP | Fixed | 13.00 | 0.34 [0.33, 0.36] | 40.43 | 96.66 | | | | |
| | Random | 13.00 | 0.32 [0.22, 0.42] | 6.02 | | 5002 | 33 | 0 | |
| SAV | Fixed | 12.00 | 0.17 [0.14, 0.19] | 13.52 | 92.30 | | | | |
| | Random | 12.00 | 0.16 [0.07, 0.25] | 3.45 | | 698 | 9 | 0 | |
| OAP | Fixed | 12.00 | 0.15 [0.13, 0.17] | 12.30 | 91.61 | | | | |
| | Random | 12.00 | 0.14 [0.05, 0.22] | 3.13 | | 544 | 6 | 3 L | 0.07 [−0.02, 0.17] |
| OAV | Fixed | 12.00 | 0.12 [0.09, 0.14] | 9.64 | 94.00 | | | | |
| | Random | 12.00 | 0.11 [0.01, 0.20] | 2.05 | | 329 | 3 | 0 | |
| | | | | Desired Motivations | | | | | |
| TAP | Fixed | 15.00 | 0.42 [0.41, 0.43] | 63.77 | 92.07 | | | | |
| | Random | 15.00 | 0.39 [0.35, 0.44] | 14.93 | | 5343 | 53 | 0 | |
| TAV | Fixed | 14.00 | 0.29 [0.28, 0.30] | 42.36 | 92.79 | | | | |
| | Random | 14.00 | 0.26 [0.21, 0.31] | 9.18 | | 6311 | 28 | 1 L | 0.25 [0.20, 0.30] |
| SAP | Fixed | 15.00 | 0.44 [0.43, 0.45] | 67.33 | 95.09 | | | | |
| | Random | 15.00 | 0.40 [0.34, 0.45] | 11.75 | | 6723 | 56 | 0 | 0.39 [0.33, 0.44] |
| SAV | Fixed | 14.00 | 0.29 [0.28, 0.30] | 42.46 | 94.33 | | | | |
| | Random | 14.00 | 0.25 [0.19, 0.30] | 7.78 | | 5991 | 28 | 1 R | 0.27 [0.20, 0.33] |
| OAP | Fixed | 15.00 | 0.20 [0.19, 0.21] | 28.63 | 92.28 | | | | |
| | Random | 15.00 | 0.20 [0.14, 0.25] | 7.20 | | 3364 | 16 | 0 | |
| OAV | Fixed | 14.00 | 0.20 [0.19, 0.22] | 29.06 | 94.17 | | | | |
| | Random | 14.00 | 0.18 [0.11, 0.24] | 5.60 | | 2851 | 15 | 0 | |
| | | | | Positive Emotions | | | | | |
| TAP | Fixed | 15.00 | 0.28 [0.26, 0.29] | 31.56 | 94.39 | | | | |
| | Random | 15.00 | 0.29 [0.22, 0.36] | 7.64 | | 5196 | 28 | 0 | 0.29 [0.22, 0.36] |
| TAV | Fixed | 15.00 | 0.20 [0.18, 0.22] | 22.44 | 96.19 | | | | |
| | Random | 15.00 | 0.20 [0.11, 0.29] | 4.26 | | 2479 | 16 | 6 R | 0.31 [0.21, 0.40] |
| SAP | Fixed | 14.00 | 0.27 [0.25, 0.28] | 29.79 | 92.30 | | | | |
| | Random | 14.00 | 0.26 [0.20, 0.32] | 7.90 | | 4018 | 24 | 1 L | 0.25 [0.19, 0.31] |
| SAV | Fixed | 14.00 | 0.17 [0.15, 0.19] | 18.90 | 93.20 | | | | |
| | Random | 14.00 | 0.15 [0.08, 0.22] | 4.18 | | 1421 | 11 | 0 | 0.15 [0.08, 0.22] |
| OAP | Fixed | 15.00 | 0.17 [0.15, 0.18] | 18.69 | 85.77 | | | | |
| | Random | 15.00 | 0.18 [0.13, 0.23] | 7.27 | | 1871 | 11 | 2 L | 0.16 [0.11, 0.21] |
| OAV | Fixed | 15.00 | 0.13 [0.11, 0.14] | 14.16 | 89.95 | | | | |
| | Random | 15.00 | 0.11 [0.05, 0.17] | 3.75 | | 833 | 4 | 2 L | 0.09 [0.03, 0.15] |
| | | | | Negative Emotions | | | | | |
| TAP | Fixed | 9.00 | −0.06 [−0.09, −0.04] | −5.47 | 89.92 | | | | |
| | Random | 9.00 | −0.05 [−0.12, 0.03] | −1.23 | | 54 | 0 | 0 | −0.05 [−0.12, 0.03] |
| TAV | Fixed | 9.00 | 0.03 [0.00, 0.05] | 2.27 | 73.40 | | | | |
| | Random | 9.00 | 0.02 [−0.03, 0.06] | 0.64 | | 0 | 0 | 1 L | 0.01 [−0.04, 0.05] |
| SAP | Fixed | 8.00 | −0.05 [−0.06, −0.03] | −4.48 | 88.12 | | | | |
| | Random | 8.00 | −0.04 [−0.12, 0.03] | −1.22 | | 36 | 0 | 1 R | −0.02 [−0.10, 0.05] |
| SAV | Fixed | 8.00 | 0.04 [0.02, 0.07] | 3.75 | 61.94 | | | | |
| | Random | 8.00 | 0.03 [−0.01, 0.07] | 1.36 | | 12 | 0 | 1 R | 0.03 [−0.00, 0.07] |
| OAP | Fixed | 10.00 | 0.05 [0.03, 0.07] | 4.53 | 81.68 | | | | |
| | Random | 10.00 | 0.05 [0.00, 0.11] | 1.84 | | 60 | 0 | 0 | 0.05 [0.00, 0.11] |
| OAV | Fixed | 10.00 | 0.09 [0.07, 0.11] | 7.87 | 11.58 | | | | |
| | Random | 10.00 | 0.09 [0.06, 0.11] | 7.10 | | 176 | 0 | 1 R | 0.10 [0.07, 0.13] |
| | | | | Performance | | | | | |
| TAP | Fixed | 4.00 | 0.16 [−0.03, 0.06] | 8.13 | 82.74 | | | | |
| | Random | 4.00 | 0.19 [−0.08, −0.03] | 3.42 | | 100 | 3 | 1 R | 0.21 [0.11, 0.13] |
| TAV | Fixed | 4.00 | 0.03 [−0.12, 0.03] | 1.29 | 9.61 | | | | |
| | Random | 4.00 | 0.03 [0.02, 0.07] | 1.33 | | 1 | 0 | 2 L | 0.13 [−0.03, 0.06] |
| SAP | Fixed | 4.00 | 0.13 [−0.01, 0.07] | 6.32 | 94.58 | | | | |
| | Random | 4.00 | 0.09 [0.03, 0.07] | 0.93 | | 36 | 2 | 0 | 0.09 [0.03, 0.07] |
| SAV | Fixed | 4.00 | 0.00 [0.00, 0.11] | 0.10 | 74.16 | | | | |
| | Random | 4.00 | 0.00 [−0.09, 0.09] | −0.08 | | 0 | 0 | 1 R | 0.04 [−0.07, 0.14] |
| OAP | Fixed | 7.00 | 0.12 [0.09, 0.15] | 8.06 | 89.57 | | | | |
| | Random | 7.00 | 0.15 [0.05, 0.25] | 2.86 | | 168 | 2 | 1 R | 0.17 [0.07, 0.26] |
| OAV | Fixed | 5.00 | 0.00 [−0.04, 0.03] | −0.21 | 72.43 | | | | |
| | Random | 5.00 | 0.01 [−0.08, 0.10] | 0.24 | | 0 | 0 | 2 L | −0.04 [−0.14, 0.05] |

Abbreviations: TAP = task approach, TAV = task avoidance, SAP = self-approach, SAV = self-avoidance, OAP = other approach, OAV = other avoidance. Note: *p*-values all <0.001 for Z-values greater than; *k* = number of samples, *SE* = standard error, 95% LL = 95% confidence interval lower limit, 95% UL = 95% confidence interval upper limit, CI = confidence interval.

### 3.6. Certainty of Results

Table 10 contains our hypotheses, the basis for each certainty rating, and the certainty rating.

**Table 10.** Certainty of results.

| Research Hypothesis (H) | Certainty Rating and Basis |
|---|---|
| H1: Our first hypothesis concerned the overall pattern of achievement goal endorsement, the intercorrelations among the achievement goals, and reliability of the used measures. We hypothesized that (1a) participants will endorse the task and self-approach goals more than the task- and self-avoidance goals and the other avoidance goal endorsement will be greater or equivocal to other approach goal endorsement; and (1b) = participants will endorse the task and self-goals more than the other goals; (2) the range of intercorrelations will be moderate in meaningfulness; and (3) the reliability coefficient average will be acceptable. | High: Mean data with corresponding 95% confidence intervals verify that participants endorsed the task- and self-approach goals more than task- and self-avoidance goals. This pattern was reversed with the other goals, and the task goals endorsed more than self-goals, both of which endorsed more than other goals. High: The pattern of intercorrelations suggests different goals (i.e., largest correlation 0.66). High: The various achievement goal measures are reliable based on the averaged and 95% confidence intervals. |
| H2: Our second hypothesis concerned differences in the overall pattern of achievement goal endorsement based on domain and the compulsory nature of domain. We hypothesized overall greater goal endorsement in sport than education and in non-compulsory activities than compulsory ones, with the differences being more pronounced in the compulsory analyses. We hypothesized other avoidance goals to be greater in PE than sport. | High: Goal endorsement is greater in sport and non-compulsory activities than in education and compulsory activities. Significant differences emerged and effect size values ranged from large to very large. High: Differences were more pronounced in the compulsory analyses than the sport/education analyses based on Hedges' *g* values. None: We were unable to assess whether PE and sport differed in other avoidance goals. |
| H3: Our third hypothesis concerned the relationships between correlates. We hypothesized the approach goals to be most related to our outcomes compared to the avoidance goals. We expected most relationships to be small in meaningfulness. | Low: Approach goals were more related in three instances (i.e., facilitative learning strategies, desired motivations, and performance), but not for our emotion analyses. Medium: Most pronounced differences for self-goals (facilitative learning strategies, desired motivations, and positive emotions) and for the task goals (desired motivations and positive emotions). Other goal differences found for performance. High: Magnitude of correlations in line with hypothesis. |

## 4. Discussion

This quantitative review is the first systematic review with meta-analysis of the $3 \times 2$ achievement goals. Hence, we aimed to use meta-analytic techniques to summarize the state of the $3 \times 2$ achievement goals across education, sport, and occupation literatures. To achieve our overall aim, we examined three hypotheses ranging from the overall pattern of $3 \times 2$ achievement goal endorsement, to selected potential moderators, and lastly, to the relationships with correlates in the peer-reviewed literature. We located 56 studies, with the majority being in education, from primary to higher education. Of the 44 education studies, 13 investigated the achievement goals within a PE context. In traditional sport and physical activity meta-analyses (e.g., [14,15]), PE is included. We examined PE in the education domain. Though with limitations discussed later, our review provides a comprehensive review of state of the $3 \times 2$ from 2011, the first publication, until the end of our search in early 2023.

Our first hypothesis concerned the overall pattern of achievement goal endorsement, the intercorrelations among the achievement goals, and the reliability of the $3 \times 2$ measures. Our results verified our hypothesis and confirmed Elliot and colleagues' [9] initial findings that participants endorsed approach goals more than avoidance goals and endorsed task-more than self-goals, both of which participants were endorsed more than other goals. We addressed two moderators as to why participants endorsed the other-avoidance goal more than the other-approach goal in our second aim. In addition to our planned analysis, it could be that socioeconomic status and culture (independent, interdependent) impacted our results. Lochbaum et al. [15] report supports that countries with lower socioeconomic and interdependent conditions endorsed the performance avoidance goals more than countries with higher socioeconomic and independent conditions did. Lochbaum et al. included 116 articles. With 56 (providing 58 samples) articles across three domains, we were unable to provide adequate samples of socioeconomics and culture. Regardless, the consistency of Elliot and colleagues' [9] $3 \times 2$ measure seems remarkable across 56 studies with two different response sets (5- and 7-point) and with a number of translated measures. Though always lower than the task- and self-goals, the concern remains related to other avoidance

goals, that is, to avoid doing worse than others is more endorsed than doing better than others. We sought in our second hypothesis to examine other goals in more depth.

With our second hypothesis, we attempted to understand whether domain and the compulsory nature of the domain or sub-domains such a primary, secondary, and high education moderated goal endorsement patterns. We first compared education and sport domains. Across all goals, participants endorsed all six achievement goals more in sport than in education with large in meaningfulness differences for the task and other approach goals. The mindset of completing the task (e.g., swinging a golf club in the correct manner) and doing better than others (e.g., running faster in a race) are both important in sport. Our findings provide a glimpse into the other goal pattern, in that in education but not sport, the participants endorsed the other avoidance goals more than the other approach goals ($g$ = 0.20). Within the sport samples, the difference between the other approach and avoidance goals was meaningful ($g$ = 0.76).

Our second planned moderator analysis compared the compulsory nature of the domains and sub-domains. We classified sport and higher education as not compulsory and all other education sub-domains as compulsory. We excluded the work samples as the compulsory nature of the work was unknown. The overall pattern of results indicated participants in non-compulsory activities endorsed all goals more than participants in compulsory activities. Differences were large to even very large in meaningfulness with all being at least medium-sized effect size values. Hence, compulsory activities dampen goal pursuits, and this is logical as the pursuit is mandatory. As with our education compared to sport moderator analysis, our findings provide another glimpse into other approach and avoidance goal differences. Participants in non-compulsory activities endorsed the other approach goal slightly more than the other avoidance goal ($g$ = 0.07), whereas participants in compulsory activities endorsed the other avoidance goal more than the other approach goal ($g$ = 0.37). The importance of achievement goals is salient when examining relationships with correlates that help us understand how each goal impacts achievement related pursuits. We examined the correlates in our third aim.

Our last hypothesis concerned the relationships between the correlates we extracted. We categorized the extracted correlates into five categories. Our aim was to find sufficient samples ($k$ = 10) before forming a category [29]. We found sufficient samples for learning strategies, motivation, and positive and negative emotions. We did not find sufficient samples for antecedents such as entity and incremental theories of intelligence, and subcategory correlates such as external regulation. Given performance is a valued outcome, we provided those results with insufficient samples. As hypothesized, most relationships were significant, yet small in meaningfulness whether with all approach and avoidance goals merged or separated by each of the 3 × 2 achievement goals. Negligible correlates emerged for all examined relationships and our negative emotion category. However, and important, medium-sized correlations emerged for self-goals with facilitative learning strategies, desired motivations, and positive emotions and for the task goals desired motivations and positive emotions.

Though not with the 3 × 2 achievement goals, our pattern of results is consistent with past achievement goal correlate meta-analyses [10–15]. Of note, we found a contradiction to Elliot and colleagues' [9] correlate pattern. Elliot et al. justified the separation of task and self-based goals in their antecedent analyses by demonstrating that self-approach goals have little to no relationship with intrinsic motivation, while task approach goals are related to intrinsic motivation (see [9], p. 640). We found both task- and self-approach goals to be related and almost identical to our desired motivation correlates. Of course, our desired motivation correlate category consisted of many types or motivations, not just intrinsic motivation. Even so, our finding is one of importance and casts some speculation as to whether task and self-goals are as different as Elliot and colleagues justified.

*Limitations and Future Directions*

A general limitation in the existing 3 × 2 literature includes a lack of random participant selection (see study quality ratings). An additional limitation is the variation of 3 × 2 scale names across studies measuring the same goal constructs, especially as domain varies. This variation in scale name could pose an issue in collective comparison of the six goal pursuits. An additional limitation in the literature is the inconsistent Likert scale response sets used in measuring 3 × 2 goals in study participants. The most commonly used Likert scale in our 56 included articles was the 7-point scale, with one main issue being researchers using the 7-point scale with 10 sport samples (though this only occurred with two physical education samples). Within our included articles, we found the 5-point Likert scale in the PE samples. Hence, we could not compare physical education and sport adequately, which poses a major limitation. A final limitation of the literature is the inconsistencies of dimension and valence of the 3 × 2 model measured in the included studies, which is representative of the existing literature. Of the 56 studies, 37 reported data for all six goal types that led to inadequate data for comparison between the domains in the 3 × 2 components. Concerning our study, we believe one limitation might lie in the literature search language criteria. We did not exclude studies based on language; however, search terms for qualifying literature included only English or numerical terms. This may have limited the global representation of existing 3 × 2 literature; however, Z.K. filled the gaps of relevant Turkish literature, contributing nine articles to the education domain. For other non-English 3 × 2 literature, including Spanish, Russian, and Chinese, we used Google Translate, which may have contributed to misinterpretations.

Notwithstanding the mentioned literature and study limitations, the present systematic review with meta-analysis advanced the 3 × 2 achievement goal model literature. Based on the findings and limitations, we suggest the following six future research directions.

1. All literature should use the 7-point Likert scale as described and designed by Elliot and colleagues [9]. Inconsistent response sets unnecessary limits on article-to-article comparisons as well as quantitative reviews such as ours.
2. Task and self-goals need more study to understand whether they both contribute to the 3 × 2 achievement goal model or are one and the same.
3. A line of research with performance outcomes will advance the literature. The literature in both education (e.g., [10]) and sport [13] contains performance studies, whereas the 3 × 2 achievement goal literature, based on our search and data extraction, contains few. Performance in achieving contexts is the gold standard in sports [80] and in education academic achievement (i.e., grades), it is the criteria to move forward grade by grade. In work, including professions such as sales, the outcome is of significant importance, just as it is in education and sport.
4. We believe future research on a global scale to best understand the importance of the 3 × 2 achievement goal model is valuable as the articles we found were unequal across the continents, though this was not a surprise. Articles meeting our inclusion criteria came from four of the six continents with inhabitants: North America (only from the USA), Europe, Asia, and Africa (Kenya and South Africa). Guo and colleagues [1] acquired antecedents and consequences of the mastery approach goals across 77 countries. Though a massive undertaking for one researcher, to a team of researchers within a global data collection system, the process appears to be manageable. Guo et al. reported strong cross-cultural support for their findings. Based on our data, the approach goals seem appropriate for global study.
5. To enrich future research using other-based achievement goals, Tan and colleagues' [81] research with goal complexes deserves mention as our other-based goal results were minimal. Using the other-based goals with motives such as hope for success and fear of failure, Tan et al.'s results demonstrated more meaningful relationships as compared to our correlates. For instance, with the other-approach and hope for success goal complex and positive emotions, Tan reported a correlation of 0.51, whereas in our meta-analytic findings, all other-based correlations were 0.20 or below.

6. Our last future direction concerns Elliot and colleagues' [82] potential-based achievement goals. Potential-based goals are the future trajectory of past self-based goals. Research grounded in the 3 × 2 achievement goal model using the potential-based goals would further our understanding of the value of self-based goals as predictors of valued correlates such as engagement, emotions, and performance across the academic, sport, and occupation domains.

## 5. Evidence-Based Suggestions and Conclusions

Concerning practical or evidence-based suggestions, we recommend that teachers, coaches, and managers direct students, athletes, and employees toward task- and self-based goals as these two achievement goals are related with desired learning strategies, desired motivation, positive emotions, and performance. Given within education and compulsory settings, individual achievement goals' endorsement is less frequent than in sport and non-compulsory activities; thus, teachers should promote task- as opposed to other-focused intrapersonal improvement standards. For instance, in a primary school setting, students could gauge their educational outcomes by attempts to fulfil the task (e.g., a chemistry experiment) requirements leading to greater task-approach goal endorsement thus motivation. In conclusion, the 3 × 2 achievement goals literature is diverse and furthers the achievement motivation literature. The consistency in scale scores from Elliot and colleagues' [9] first measure is remarkable. In contrast, we found the task- and self-approach scales to correlations with motivation correlates to differ from that of Elliot et al.'s findings. The importance of this difference is up for speculation and future research with potential-based goals may help elucidate and differentiate the differences between these two goals.

**Supplementary Materials:** The following supporting information can be downloaded at: https://www.mdpi.com/article/10.3390/ejihpe13070085/s1.

**Author Contributions:** Conceptualization, M.L. and C.S.; methodology, M.L., C.S. and Z.K.; formal analysis, M.L. and C.S.; resources, M.L.; data curation, M.L., C.S. and Z.K.; writing—original draft preparation, M.L. and C.S.; writing—review and editing, M.L. and C.S.; supervision, M.L. All authors have read and agreed to the published version of the manuscript.

**Funding:** This research received no external funding.

**Institutional Review Board Statement:** Not applicable for studies not involving humans or animals.

**Informed Consent Statement:** Not applicable for archival data.

**Data Availability Statement:** All analyzed data are found within this article and the corresponding Supplementary Materials.

**Acknowledgments:** We acknowledge the Department of Kinesiology and Sport Management for purchasing the Comprehensive Meta-Analysis license.

**Conflicts of Interest:** The authors declare no conflict of interest.

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
