# Peer review of "The 3 × 2 Achievement Goals in the Education, Sport, and Occupation Literatures: A Systematic Review with Meta-Analysis"

_ejihpe, doi:10.3390/ejihpe13070085_

Round 1
Reviewer 1 Report
The paper is clearly written and based on a sound research design, which is - as far as I can note - consistently and rigorously implemented. There are some limitations based on the research design, but the authors seem to be very well aware of them.
I have two minor suggestions:
1) Saying that a question concerns something is not the same as formulating a question. The authors focus on three research questions but they do not really formulate them as questions per se. They seem more like sets of related hypotheses. They may be either explicitly formulated as questions or presented as thematic sets of related hypotheses.
2) The interpretative part of the Discussion section is well-written, but not very rich in the actual in-depth interpretation of the obtained results. The authors could expand on that - both referring to existing theory and research and to their own substantiated claims.
There is also an error in page 568, where South Africa is mentioned as being within South America. In addition, the key issue the authors might emphasise here, is not about the number of included continents (Australia is not really that relevant due to its comparable size, and mentioning Antarctica is just a waste of space) but more about the clear disproportions in terms of how much different parts of the world are included. Of course, this is hardly an exception from social science research in general.
No comments. English is of sufficient quality.
Reviewer 2 Report
Congratulations to the authors for the topic under study. However, this review study presents some issues that may need to be addressed. In the comments below, some considerations are set out.
Overall, the introduction is well organized. A framework of the problem is presented as well as the definition of key concepts within the study area. However, it is suggested to specify what the authors consider as "sufficient data for analyses in at least one of our research questions." page 4, line 145.
It is also suggested to specify the range of years of publication of the studies (although this is reported in the supplemental file materials), as well as the research design of the studies to be included. It would also be useful to state what the intended age range of study participants was.
It is recommended that all stages of the selection of studies included in this systematic literature review could be clarified. How were discrepancies between researchers about the inclusion or exclusion of studies in the report resolved?
The "Dicussion" section is well structured. However, it is suggested that some practical implications (evidence-based practice), from the analyses carried out, could be better specified.
Reference number 64 is not formatted according to the journal's standards
